

**Spatio-temporal variations in surface Marine Carbonate**
**System properties across the Western Mediterranean Sea**
**using Volunteer Observing Ship data.**
David Curbelo-Hernández*, David González-Santana, Aridane González-González, J.
Magdalena Santana-Casiano and Melchor González-Dávila
[1] Instituto de Oceanografía y Cambio Global (IOCAG), Universidad de Las Palmas de
Gran Canaria (ULPGC). Las Palmas de Gran Canaria, Spain.
* Corresponding Author: david.curbelo@ulpgc.es



**Abstract**
The surface physical and Marine Carbonate System (MCS) properties were assessed
along the western boundary of the Mediterranean Sea. An unprecedent high-resolution
observation-based dataset spanning 5 years (2019-2024) was built through automatically
underway monitoring by a Volunteer Observing Ship (VOS). The MCS dynamics were
strongly modulated by physical-biological coupling dependent on the upper-layer
circulation and mesoscale features. On a seasonal scale, the variations in $CO_2$ fugacity
($f\text{CO}_{2,sw}$) were mainly driven by sea surface temperature (SST) fluctuations (45-83%) and
partially offset by the processes controlling total inorganic carbon ($C_T$) distribution (25-
38%). On an interannual scale, the SST trends (0.26-0.43 °C yr$^{-1}$) have accelerated by 78-
88% in comparison with previous decades. The ongoing surface warming was the main
factor (with a contribution of ~76-92%) increasing $f\text{CO}_{2,sw}$ (4.18 to 5.53 µatm yr$^{-1}$) and,
consequently, decreasing pH (-0.005 to -0.007 units yr$^{-1}$) in the surface waters. The
seasonal SST, becoming larger due to progressively warmer summers, was the primary
driver of the observed slope up of interannual trends. The evaluation of the air-sea $CO_2$
exchange shows the area across the Alboran Sea (14,000 Km$^2$) and the eastern Iberian
margin (40,000 Km$^2$) acting as an atmospheric $CO_2$ sink of -1.57 ± 0.49 mol m$^{-2}$ yr$^{-1}$ (0.97
± 0.30 Tg $CO_2$ yr$^{-1}$) and -0.70 ± 0.54 mol m$^{-2}$ yr$^{-1}$ (-1.22 ± 0.95 Tg $CO_2$ yr$^{-1}$), respectively.
The net annual $CO_2$ sink has reduced by 40-80% since 2019 due to the ongoing strength
of the source status during summer and the weakening in the sink status during spring and
autumn.
**Keywords:** Marine Carbonate System, Air-sea $CO_2$ fluxes, Volunteer Observing Ships,
Western Mediterranean Sea, ocean acidification, sea-surface warming





## 1. Introduction

The semi-enclosed and marginal seas have a relevant role in the global biogeochemical cycles and are highly vulnerable to climate change (IPCC, 2023). These regions accomplish extensive coastal and continental shelf and slope areas occupied with multiple diverse ecosystems under anthropogenic pressure. Although these regions present enhanced biogeochemical activity and intensified air-sea $CO_2$ exchange rates compared to the open ocean (Borges et al., 2005; Cai et al., 2006; Frankignoulle and Borges, 2001; Shadwick et al., 2010), its poorly monitoring and assessment have historically excluded them from global studies and models and underestimated in the Global Carbon Budget (Friedlingstein et al., 2023)

The Mediterranean Sea is a dynamic semi-enclosed system potentially fragile to natural and anthropogenic forcing (e. g. Álvarez et al., 2014; Tanhua et al., 2013). The particular oceanography of the Mediterranean Sea, collectively described in several works (e.g. Nielsen, 1912; Robinson et al., 2001; Millot and Taupier-Letage, 2005; Bergamasco and Malanotte-Rizzoli, 2010; Schroeder et al., 2012), have rendered it a "miniature ocean" considered as "laboratory basin" to evaluate physico-chemical perturbations that can be extrapolated to larger scales in the global ocean (e.g. Robinson and Golnaraghi, 1994; Bergamasco and Malanotte-Rizzoli, 2010). These perturbations have accelerated since the second half of the 20th century, with temperature and salinity increasing at unprecedent rates of 0.04ºC and 0.015 per decade, respectively (Borghini et al., 2014), impacting the Marine Carbonate System (MCS). However, the availability of high-quality observation-based data and research in this basin is scarce due to spatial and temporal limitations in the monitoring and sampling techniques (Millero et al., 1979; Rivaro et al., 2010).

The MCS dynamics has been evaluated in the Northwestern Mediterranean basin (Bégovic and Copin-Montégut, 2002; Copin-Montégut and Bégovic, 2002, 2004; Coppola et al., 2020; Hood and Merlivat, 2001; Mémery et al., 2002; Merlivat et al., 2018; Touratier and Goyet, 2009; Ulses et al., 2023), mainly conducted at the time-series DYFAMED (43.42 ºN, 7.87 ºE; Marty, 2002) and BOUSSOLE sites (43.37° N, 7.90° E; Antoine et al., 2006, 2008a, 2008b). These investigations have shown the seasonal cycle of the surface $CO_2$ is primarily governed by thermal fluctuations and the behaviour of the area as a relatively weak sink for atmospheric $CO_2$ on an annual scale. Long-term changes



estimated by Merlivat et al., (2018) reported the increase in the surface $CO_2$ fugacity
($f$CO$_{2,sw}$) and pH of ~40 µatm and ~0.04 units, respectively, since the 90s. The interannual
trends given for $f$CO$_{2,sw}$ ($2.3 \pm 0.23$ µatm yr$^{-1}$; Merlivat et al., 2018) and pH (0.002-0.003
units yr$^{-1}$; Yao et al., 2016) were in agreement with those encountered in the Northeast
Atlantic at the ESTOC site ($2.1 \pm 0.1$ µatm yr$^{-1}$ and $0.002 \pm 0.0001$ units yr$^{-1}$,
respectively; González-Dávila and Santana-Casiano, 2023). Although the Northwestern
Mediterranean is characterized by a relatively strong atmospheric $CO_2$ uptake and storage
due to deep-convection (Copin-Montégut, 1993; D'Ortenzio et al., 2008; Cossarini et al.,
2021), the long-term variations in MCS occur at rates larger than the expected from the
chemical equilibrium with the atmospheric $CO_2$. It has been attributed to the substantial
input of anthropogenic carbon from the North Atlantic (Merlivat et al., 2018; Palmiéri et
al., 2015; Schneider et al., 2010; Ulses et al., 2023). Based on a high-resolution regional
model, Palmiéri et al., (2015) estimated that ~25% of the anthropogenic carbon storage
in the Mediterranean Sea comes from the Atlantic. The water exchange processes in the
Strait of Gibraltar become the western boundary of the Mediterranean Sea in a crucial
region for MCS variability which significantly modulates the basin-wide anthropogenic
carbon inventory and ocean acidification trends in the Mediterranean basin and could
affect significantly the general circulation and the composition of seawaters in the North
Atlantic. Additionally, this region is subject to variability related with (1) the intense
deep-water convection in the adjacent Northwestern area of the Mediterranean Sea and
(2) the unique circulation patterns shaped to the irregular coastlines and islands, which
forms quasi-permanent eddies and other (sub)mesoscale features (Alberola et al., 1995;
Bosse et al., 2021; 2016; Bourg and Molcard, 2021).
The Western Mediterranean Sea encompasses the Alboran Sea, land-loaded by the
southern Iberian Peninsula coast and northern African coast, and the coastal transitional
area along the eastern Iberian margin (Figure 1a). The classical surface circulation pattern
in the Alboran Sea (e. g. Bormans and Garrett, 1989; Peliz et al., 2013; Sánchez-Garrido
et al., 2013, 2022; Speich, 1996; Whitehead and Miller, 1979), with the Atlantic water jet
(AJ) following wavelike path of the quasi-permanent Western Anticyclonic Gyre (WAG)
and the Eastern Anticyclonic Gyre (EAG) and constituting the Modified Atlantic Water
(MAW; Lopez-García et al., 1994; Viúdez et al., 1998), drive west-to-east variations in
physical and biogeochemical terms. The intensity and direction of the AJ, depending
primarily on sea level pressure and local wind fluctuations, variate on different timescales



and govern the circulation patterns in the Alboran Sea influencing the biogeochemistry
(Sánchez-Garrido and Nadal, 2022; Solé et al., 2016). On a seasonal scale, the AJ oscillate
between two main circulation modes (García-Lafuente et al., 2002; Macías et al., 2008,
2016; Vargas-Yáez et al., 2002), detectable by reanalysis data-based SST signals (Figure
1b): a high-intense AJ flowing north-eastward during spring/summer and a lower-intense
AJ flowing with more south-eastwardly direction during autumn/winter. The stronger AJ
during the warm months feed the classical two-gyres configuration in the Alboran Sea,
while the weak AJ only allows the exitance of the WAG (Renault et al., 2012). The AJ
forms a filament flowing from the Iberian coastal upwelling in the northwestern Alboran
Sea and surrounding the eastern edge of the WAG, which is most frequently presented
during summer (Gómez-jakobsen et al., 2019; Millot, 1999). The westernmost part of the
Alboran Sea is affected by the shallow position of the Atlantic-Meridional Interface layer
(AMI; Bray et al., 1995; Lacombe and Richez, 1982), which promotes the injection of
deep-water into the surface (Echevarría et al., 2002; Gómez-jakobsen et al., 2019; Minas
et al., 1991).
The eastern Iberian margin is influenced by the path of the Northern Current transporting
Mediterranean Water (MW; Pinot et al., 1995), which is originated around the Gulf of
Lion where the forcing of the northeasterly winds is frequently strong and flows
southward along the eastern coastline of the Iberian Peninsula (Conan and Millot, 1995;
Millot, 1999; Sammari et al., 1995). The seasonality of the Northern Current (Millot,
1999) infers meridional variations in the thermal signals between cold and warm months
(Figure 1b). The enhanced wind-forcing during winter intensify the Northern Current,
which fit to the Iberian continental slope and recirculate offshore at Cape of Nao, while a
low-intense branch progress southward Cape of Nao and reach the eastern Alboran Sea.
The weakening in the wind-forcing forms a surface thermal front in the axis of the
Pyrenees during summer and changed the path of the Northern Current further away from
the Iberian coast (Lopez-García et al., 1994), which allow the MAW to reach its northern
most spreading.
This research focus on the surface spatio-temporal variations of the MCS and air-sea $CO_2$
fluxes in the western boundary of the Mediterranean Sea. An alternatively and efficiently
observation-based method that ensures high-frequency and quality data was used: the
autonomous underway monitoring of the surface ocean by a Volunteer Observing ship





(VOS). This systematic strategy represents a powerful tool to analyse the distribution and
changes of physical and MCS properties in highly variable areas as coastal transitional
zones where the availability of data has been historically scarce. The dataset used was
built based on continuous observations along the SOOP CanOA-VOS line (Curbelo-
Hernández et al., 2021a; 2021b) from February 2019 to February 2024. The cruise track
(Figure 1) followed the south and east geographically rugged coastline of the Iberian
Peninsula and allowed the characterization of the Alboran Sea (~2-5.1ºW) separately
from the eastern coastal and shelf area between Cape of Gata (Almería) and Barcelona
(~36.5-41.3ºN). The changes observed in the MCS on a seasonal and interannual
timescales (even considering the limitations of 5 years of data), the mechanism
controlling their variations and the changes in the air-sea $CO_2$ exchange have been
attended in this study, contributing to improve our knowledge in a key oceanographic
region.

## 2. Material and methods

### 2.1. Data collection

A high spatio-temporal resolution dataset spanning 5 years was constructed based on
weekly physico-chemical observations of the surface western boundary of the
Mediterranean Sea between February 2019 and February 2024. Data was automatically
collected by a Surface Ocean Observation Platform (SOOP) running in underway mode
and placed aboard the Volunteer Observing Ship (VOS) MV JONA SOPHIE (IMO:
9144718, called RENATE P before November 2021), a container ship managed in Spain
by Nisa Maritima which links the Canary Islands with Barcelona.
The SOOP CanOA-VOS line allows the monitoring of the northeast archipelagic waters
of the Canary Islands and coastal transitional waters of the Northeast Atlantic (Curbelo-
Hernández et al., 2021), the Strait of Gibraltar (Curbelo-Hernández et al., 2021) and the
western Mediterranean Sea (Figure 1). The system operates fully unattended with
biweekly (time required to complete a round trip) routine maintenance at the port of Las
Palmas de Gran Canaria (28.13 ºN, 15.42 ºW). The automatic transfer of data to a server
occurs each time the vessel docks at each of the port along the usual route (Las Palmas
de Gran Canaria, Santa Cruz de Tenerife, Arrecife, Sagunto and Barcelona). A total of 92
routes were completed in the Mediterranean Sea (Figure 1).





The SOOP CanOA-VOS line, which was designed and is maintained by the QUIMA
research group at the IOCAG-ULPGC, is part of the Spanish contribution to the
Integrated Carbon Observation System (ICOS-ERIC; https://www.icos-cp.eu/) since
2021 and has been recognized as an ICOS Class 1 Ocean Station. Therefore, the
measurement equipment and underway data collection techniques verify the ICOS-ERIC
high-quality requirements and methodological recommendations.

### 2.2. Monitoring routines

The autonomous underway monitoring of $CO_2$ in surface ocean and low atmosphere and
the data collection routines followed the recommendations described by Pierrot et al.,
(2009) to ensure comparable and high-quality datasets. An automated underway $CO_2$
molar fraction ($xCO_2$, ppm) measurement system, developed by Craig Nail and
commercialized by General Oceanics™, was installed inside the engine room of the
SOOP CanOA-VOS and described in detail by Curbelo et al. (2021a, 2021b).
The $xCO_2$ measurement system combines an air and seawater equilibrator, placed inside
the wet box, with a non-dispersive infrared analyser for gas detection, placed inside the
dry box. The analyser used for $xCO_2$ detection was built by LICOR® (initially the 6262
model and after October 2019, a 7000 model). The analyser is automatically calibrated
on departure and arrival at each port and periodically in loop every three hours using four
standard gases. Additionally, the system is zeroed and spanned (with standard gases 1 and
4, respectively) every twelve hours to properly interpolate the standard values and correct
for instrument drift. The four standard gases, with an accuracy of ±0.02 ppm, were
provided by the National Ocean and Atmospheric Administration (NOAA) and traceable
to the World Meteorological Organization (WMO). They were in the order of 0 ppm, 250
ppm, 400 ppm and 550 ppm until January 2021, when the gas bottles for standard 2 to 4
were changed for a new set with concentrations in the order of 300 ppm, 500 ppm and
800 ppm provided by the ICOS central analytical laboratories.
The sea surface temperature (SST, in ºC) was monitored by using a SBE38 thermometer
placed at the primary seawater intake in the engine room, with a reported error of ±0.01ºC.
The high sensitivity of $xCO_2$ to temperature fluctuations required to measure the
temperature at different locations along the system. A SBE45 thermosalinograph and a
Hart Scientific HT1523 Handheld Thermometer, with reported errors of ±0.01ºC, were
used to monitor the temperature at the entrance of the wet box and inside the equilibrator,



respectively. The SBE45 thermosalinograph measured the sea surface salinity (SSS) with
an estimated error of ±0.005. Lastly, the atmospheric pressure is monitored at the deck
box transducer, while the differential pressure with the ambient air is also controlled in
the wet box inside the equilibrator and in the dry box inside the analyser. The atmospheric
pressure records can differ in the order of milibars with the pressure inside the engine
room due to the forcing of ventilation.
Discrete surface seawater samples were manually collected with in situ records of SST
and SSS during three round trips in February 2020, March 2021 and October 2023. The
discrete sampling was performed along the vessel track from the seawater supply line
every 1-2 hours in borosilicate glass bottles, overfilled and preserved with 100 µl of
saturated $HgCl_2$. Samples were kept in dark and analysed just after arriving at port, in a
period less than 2 weeks, for total alkalinity ($A_T$, µmol kg$^{-1}$) and total dissolved inorganic
carbon ($C_T$, µmol kg$^{-1}$) determination A total of 102 discrete samples has been collected
in the Mediterranean Sea.
The underway observational dataset exhibits a gap of a year among September 2021 and
2022 due to the temporary cessation of the measurement system for vessel maintenance
activities in dry dock. During this period, the measurement system was sent for calibration
and maintenance to General Oceanics enterprise, Miami, USA. There are also several
gaps of less than a month related with different technical issues with the measurement
equipment, which were addressed during the routine maintenance visits to the vessel (i.
e. problems with the pump and seawater intake, with the LICOR analyser, depletion of
gas bottles supplies, electrical issues in the engine room). Certain technical issues
encountered during 2020 were delayed in being resolved due to the constraints imposed
by COVID-19.
**2.3. Calculation procedures**
**2.3.1.  CO₂ system variables**
The present investigation followed the data collection methodology, quality control and
calculation procedures as published in the updated version of the DOE method manual
for ocean $CO_2$ analysis (Dickson et al., 2007). The post-cruises correction of the measured
$xCO_2$ and calculation of the fugacity of $CO_2$ in surface seawater ($fCO_{2,sw}$) and in the lower
atmosphere ($fCO_{2,atm}$) followed the procedure described by Pierrot et al. (2009). The full





set of standard gases was linearly interpolated to the time of observations to generate the
calibration curve used for $xCO_2$ correction before calculating $fCO_2$.
The discrete seawater samples were analysed for $A_T$ and $C_T$ by using a VINDTA 3C and
following the procedure detailed by Mintrop et al., (2000).. The VINDTA 3C was
calibrated through the titration of Certified Reference Material (CRMs; provided by A.
Dickson at Scripps Institution of Oceanography), giving values with an accuracy of ±1.5
µmol kg$^{-1}$ for $A_T$ and ±1.0 µmol kg$^{-1}$ for $C_T$. The $A_T$ was calculated at the times of the
observations as previously done in the Northeast Atlantic (Curbelo-Hernández et al.,
2021; 2023) and in the Strait of Gibraltar (Curbelo-Hernández et al., 2021), using the $A_T$-
SSS linear relationship obtained from the discrete samples (Eq. 1), which is statistically
significant at the 99% level of confidence (p-value < 0.01; $r^2$= 0.92). The change in $A_T$
with SSS was assumed as constant through the entire annual cycle at this latitudes (Lee
et al., 2006). The $A_T$-SSS relationship provided here can be used to calculate the $A_T$
content of surface seawaters in the Mediterranean Sea with salinities ranging between 36
and 38.5 and with a standard error of estimate of ±17.1 µmol kg$^{-1}$ (<0.7%).
$$A_T = 101.4 \ (\pm 6.3) \ SSS - 1303 \ (\pm 234) \tag{1}$$
The pH and $C_T$ were calculated at the times of the underway observations by using the
$CO_{2SYS}$ programme developed by Lewis and Wallace, (1998) and run with the MATLAB
software (van Heuven et al., 2011; Orr et al., 2018; Sharp et al., 2023). The $fCO_{2,sw}$ and
$A_T$ were used as input $CO_2$ system variables. The set of constant used for computations
includes the carbonic acid dissociation constants of Lueker et al., (2000), the $HSO_4$
dissociation constant of Dickson, (1990), the HF dissociation constant of Perez and Fraga,
(1987) and the value of $[B]_T$ determined by Lee et al., (2010). The effect of temperature
on pH was removed by computation at a constant temperature of 19ºC, which is the mean
temperature within the observational period (referred as $pH_{19}$).

### 2.3.2.    Thermal and non-thermal $fCO_{2,sw}$

The contribution of the thermal and non-thermal processes on the variation of $fCO_{2,sw}$ has
been addressed. The non-thermal processes mainly include the biological and carbonate
pumps, circulation patterns and air-sea gas exchange (De Carlo et al., 2013). The
collectively known methodology presented by Takahashi et al., (2002) with the



experimentally-determined temperature effects on pCO$_2$ for isochemical seawater of
0.0423 ºC$^{-1}$ (Takahashi et al., 1993) was used. This procedure has been previously applied
to SOOP CanOA-VOS data and detailed by Curbelo-Hernández et al., (2021a; 2021b).
An alternative procedure recently introduced by Fassbender et al., (2022) and detailed by
Rodgers et al., (2023),modified from the Takahashi et al., (2002, 1993) framework was
also used in this investigation. This updated method addresses the slightly variations in
the thermal sensitivity of $f$CO$_{2,sw}$ due to background chemistry (Wanninkhof et al., 1999,
2022), which introduces slightly difference between the observed seasonal cycle of
$f$CO$_{2,sw}$ and the calculated through the sum of its thermal and non-thermal components.
The new approach for the thermal component of $f$CO$_{2,sw}$ ($f$CO$_{2, T FASS}$) was computed from
the annual means (denoted with the subscripts AM) of SSS, A$_T$ and C$_T$ at in situ
temperature (Eq. 2) by using the CO$_{2SYS}$ programme (Lewis and Wallace, 1998) for
MATLAB (van Heuven et al., 2011; Orr et al., 2018; Sharp et al., 2023).

$$fCO_{2,\ T\ FASS} = CO_{2,SYS}(C_{T,AM},\ A_{T,AM},\ SSS_{AM}, SST) \tag{2}$$

The thermal-driven change in $f$CO$_{2,sw}$ ($f$CO$_{2, T anom}$) can be calculated as the difference
between the thermal component of $f$CO$_{2,sw}$ ($f$CO$_{2, T FASS}$) and the annual mean of $f$CO$_{2,sw}$
(Eq. 3). The non-thermal component ($f$CO$_{2, NT FASS}$) is given by the difference between
the $f$CO$_{2,sw}$ at the times of observations and the $f$CO$_{2, T anom}$ (Eq. 4). The difference among
$f$CO$_{2, NT FASS}$ and the annual mean of $f$CO$_{2,sw}$ provides the change in $f$CO$_{2,sw}$ explained by
non-thermal processes ($f$CO$_{2, NT anom}$) (Eq. 5).

$$fCO_{2,\ T\ anom} = fCO_{2,\ T\ FASS} - fCO_{2,\ AM} \tag{3}$$

$$fCO_{2,\ NT\ FASS} = fCO_{2,sw} - fCO_{2,\ T\ anom} \tag{4}$$

$$fCO_{2,\ NT\ anom} = fCO_{2,\ NT\ FASS} - fCO_{2,\ AM} \tag{5}$$

The relative importance of thermal and non-thermal processes was expressed by the T/B
ratio ($\Delta f$CO$_{2,thermal}$/$\Delta f$CO$_{2,non-thermal}$), with values greater than 1 indicating that the
temperature effect govern the $f$CO$_{2,sw}$ variations.
**2.3.3.  Factors controlling the seasonality of $f$CO$_{2,sw}$**



The changes in the surface $f\text{CO}_{2,\text{sw}}$ result from the combined variation in the physical and biochemical seawater properties. The seasonal variability of the surface $f\text{CO}_{2,\text{sw}}$ was addressed by attending the partial contribution of SST, SSS, $C_T$ and $A_T$. The influence of each driver was quantified by assuming linearity and employing a first-order Taylor-series deconvolution (Sarmiento and Gruber, 2006) given in Eq. 6 and previously used for $p\text{CO}_2$ (Doney et al., 2009; Lovenduski et al., 2007; Takahashi et al., 1993; Turi et al., 2014) and pH (Fröb et al., 2019; García-Ibáñez et al., 2016; Pérez et al., 2021; Takahashi et al., 1993; Curbelo-Hernández et al., 2024). Due to the high relevance of the evaporation/precipitation processes in the Mediterranean Sea and in order to avoid the influence of river discharge and other freshwater fluxes along the south and east coast of the Iberian Peninsula, the most recent equation (Eq. 7) given by Pérez et al., (2021) with salinity-normalized $C_T$ and $A_T$ ($NX_T = X_T/S*37.4$) was used. The $C_T$ and $A_T$ were normalized ($NC_T$ and $NA_T$) to a constant salinity of 37.4, the average for the entire monitored area ($NX_T = X_T/SSS*37.4$).

$$\frac{dpCO_2}{dt} = \frac{\partial pCO_2}{\partial SST}\frac{dSST}{dt} + \frac{\partial pCO_2}{\partial SSS}\frac{dSSS}{dt} + \frac{\partial pCO_2}{\partial C_T}\frac{dC_T}{dt} + \frac{\partial pCO_2}{\partial A_T}\frac{dA_T}{dt} \tag{6}$$

$$\frac{dpCO_2}{dt} = \frac{\partial pCO_2}{\partial SST}\frac{dSST}{dt} + \left(\frac{\partial pCO_2}{\partial SSS} + \frac{NC_T}{SSS_0}\frac{\partial pCO_2}{\partial C_T} + \frac{NA_T}{SSS_0}\frac{\partial pCO_2}{\partial A_T}\right)\frac{dSSS}{dt} + \frac{SSS}{SSS_0}\frac{\partial pCO_2}{\partial C_T}\frac{dNC_T}{dt} + \frac{S}{S_0}\frac{\partial pCO_2}{\partial A_T}\frac{dNA_T}{dt}$$
$$\tag{7}$$

It is important to remark that the changes in $NA_T$ and $NC_T$ are linked with biogeochemical processes which have different influences: the processes involved in the organic carbon pump contribute to strongly change the $NC_T$ weakly affecting the $NA_T$, while those involved in the carbonate pump affect the $NA_T$ twice as much as $NC_T$.

### 2.3.4. Air-sea $\text{CO}_2$ fluxes

The $\text{CO}_2$ fluxes ($\text{FCO}_2$) were determined using Eq. 8 with a conversion factor of 0.24 mmol m$^{-2}$ d$^{-1}$. The solubility ($S$) and the difference between seawater and low atmosphere $f\text{CO}_2$ ($\Delta f\text{CO}_2 = f\text{CO}_{2,\text{sw}} - f\text{CO}_{2,\text{atm}}$) were considered. Negative fluxes indicate that the ocean acts as an atmospheric $\text{CO}_2$ sink, while the positive ones indicate that it behaves as a source.

$$\text{FCO}_2 = 0.24 \cdot S \cdot k \cdot \Delta f\text{CO}_2 \tag{8}$$



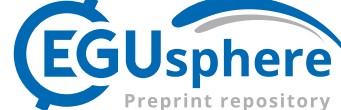

The Wanninkhof (2014) parameterization was used in this study, with $k$ being the gas
transfer rate expressed in Eq. 9:

$$k = 0.251 \cdot w^2 \cdot \left(\frac{Sc}{660}\right)^{-0.5} \tag{9}$$

where $w$ is the wind speed (m s$^{-1}$) and $Sc$ is Schmidt number (cinematic viscosity of
seawater, divided by the gas diffusion coefficient). Both $S$ and $Sc$ were calculated with
the equations and coefficients given by Wanninkhof (2014) for $CO_2$ in seawater. ERA5
hourly wind speed reanalysis data at 10 m above the sea level and with a spatial resolution
of 0.25º x 0.25º (Hersbach et al., 2023) were used to calculate $k$. The ERA5 reanalysis for
the global climate and weather is available at Copernicus Climate Data Store
(https://cds.climate.copernicus.eu/).
**2.4. Data adjustments and statistical procedures**
The raw output data was initially filtered removing data affected by the automatic sampler
such as samples measured at low water rates (< 2.0 L min$^{-1}$) and/or samples in which the
difference in temperature between the seawater intake and the equilibrator was higher
than 1.5ºC. The outliers, assumed as elements more than three local standard deviations
from the local mean over a window length of fifty elements, were also removed from the
dataset. The $xCO_2$ measured values in low atmosphere after each calibration were
averaged and interpolated at the times of each $xCO_2$ observation in seawater by applying
a piecewise polynomial-based smoothing spline.
The temporal evolution of the physico-chemical data was analysed by weekly averaging
(time required by the vessel to complete a trip) at different locations along the vessel
track. The average values (y) were fitted to Eq. 10 as a function of time (year fraction).
This equation update the one used to study seasonal cycles by Curbelo-Hernández et al.,
(2021a; 2021b) through the addition of the $b \, (year - 2019)$ term, which provides the
interannual rate of change of each seasonally-detrended variable between 2019 and 2024.
The coefficients *a-f* and the standard errors of estimate given by Eq. 10 for the variables
considered are available in Table Sup1.
$y = a + b \, (year - 2019) + c \cdot \cos(2\pi \, year) + d \cdot \sin(2\pi \, year) + e \cdot \cos(4\pi \, year) +$
$f \cdot \sin(4\pi \, year)$        (10)





The errors in the weekly averages were determined by dividing the Standard Deviation
by the square root of the number of data points used to calculate the means
($Standard\ Deviation/\sqrt{n}$). The coefficient $b$ in Eq. 10 represented the interannual
variation rates for each variable, which coincided with the slope derived from linear
regressions of the detrended average values over time. The standard errors of these slopes
were calculated by propagating the errors from the annual mean values.
To evaluate the strength and direction of the linear regressions and the significance of the
interannual trends, we applied the Pearson correlation test. This test yielded correlation
coefficients ($r^2$) and corresponding $p$-values to determine statistical significance. Trends
with $p$-values ≤ 0.01 were statistically significant at the 99% confidence level, those with
$p$-values ≤ 0.05 were significant at the 95% confidence level, and trends with p-values ≤
0.1 were significant at the 90% confidence level. Trends with $p$-values > 0.1 were not
statistically significant but still provided an estimate of the temporal evolution of the
variables within their respective layers.
## 3. Results

### 3.1. Spatial distribution of the surface physicochemical properties.
The surface underway monitoring allowed a high-resolution characterization of the
western boundary of the Mediterranean Sea. A total amount of 157,984 data for surface
ocean $xCO_2$ were collected during the study period (34,015 data during 2019, 28,590 data
during 2020, 33,288 data during 2021, 19,102 data during 2022, 39,738 data during 2023
and 3,251 data during January and February 2024). Based on differences in the spatial
distribution of the observation-based data and in the heterogeneous influence of
hydrodynamical processes and oceanographic features, two subregions (referred to as
sections) were identified along the vessel track (Figure 1): the longitudinally distributed
southern section (hereinafter S section), accomplishing the Alboran Sea (~2-5.1ºW), and
the latitudinally distributed east section (hereinafter E section), following the eastern
coastline of the Iberian Peninsula (~36.5-41.3ºN).
The spatial distribution of the average values allowed to identify heterogeneity in the
annual cycle of each variable along the longitudinal S section and latitudinal E section
(Figure 2 and Sup1). The standard deviation of the spatially-averaged variables is





presented in Table Sup2. A strong west-to-east increasing gradient in SST was observed
in summer through the S section (~5.5ºC) which lead an increment in $f$CO$_{2,sw}$ of ~57.5
µatm and a depletion in pH of ~0.040 units from the Strait of Gibraltar to the Cape of
Gata. Despite the approximately constant SST through the S section during the rest of the
year (less than 1.5ºC of difference between the western and easternmost parts), an
eastward decrease in $f$CO$_{2,sw}$ of less than 18 µatm accompanied by an increase in pH of
less than 0.030 units was observed between October and March.
The latitudinal gradient of SST through the E section was weaker throughout the year,
keeping spatially stables the $f$CO$_{2,sw}$ and pH. The maximum change in SST occurs during
winter, in which a northward decrease of less than 2ºC explained minimum seasonal
average temperatures and $f$CO$_{2,sw}$ through the cruise track (14-15 ºC and 350-360 µatm,
respectively). It contrasts with the maximum average temperatures and $f$CO$_{2,sw}$
encountered during summer (25.0-26.5 ºC and 450-470 µatm, respectively). These results
reported that the maximum amplitude of the seasonal cycle of SST, $f$CO$_{2,sw}$ and pH occurs
along the eastern coastline of the Iberian Peninsula and specially over the continental
shelf between Valencia and Barcelona (northernmost part of E section), while the
minimum seasonal amplitude occurs near the Strait of Gibraltar (westernmost part of the
S section).
The spatial variation in C$_T$ (Figure 2) were significant throughout the year along both
sections in phase with the distribution of A$_T$ and the strong gradient in SSS (Figure Sup1).
The C$_T$ increases eastward in the order of 20-45 µmol kg$^{-1}$ in the Alboran Sea throughout
the year. This increment accelerated from Cape of Gata to Cape of Nao, where the average
C$_T$ become approximately stable until Barcelona. The spatial distribution of C$_T$ and A$_T$
was highly influenced by the progressively salinification observed in the semi-enclosed
transitional area between the Strait of Gibraltar and the Mediterranean Sea. The SSS
increased during the entire annual cycle from 36.3-36.5 around the eastern part of the
Strait of Gibraltar to 37.7-38.1 around Cape of Nao (Figure Sup1). Removing the effect
of salinity, the NC$_T$ (Figure Sup1) presents a weaker spatial variation through the vessel
track mainly lead by biological and mixing processes.
**3.2. Seasonal cycle of the SST, SSS and MCS.**



The surface physico-chemical properties show heterogeneities during some seasons of
the year among several key locations along the sections (Figure 2 and Sup1). The
heterogeneities in the temporal evolution of the SST, SSS and $CO_2$ system variables was
assessed by the strategic selection of 5 stations along the S section (stations S1-S5) and 6
stations along the E section (stations E1-E6), geographically depicted in Figure 1. The S1
(4.95 ± 0.05 ºW) occupied the easternmost part of the Strait of Gibraltar, the S2-S4 (4.35
± 0.05 ºW, 3.85 ± 0.05 ºW and 2.95 ± 0.05ºW) were placed in the central Alboran Sea
and the S5 (2.45 ± 0.05 ºW) located south of Cape of Gata. The stations along the E
section include E1 (37.1 ± 0.2 ºN) in the Gulf of Mazarron, E2 (37.6 ± 0.2 ºN) to the east
of Cape of Palos, E3 (38.2 ± 0.2 ºN) in the Gulf of Alicante, E4 (38.7 ± 0.2 ºN) to the east
of Cape of Nao, E5 (39.3 ± 0.2 ºN) in the Gulf of Valencia over the continental slope, and
E6 (40.2 ± 0.2 ºN) near the Ebro estuary over the continental shelf.
The temporal variations of each variable at S1-S5 and E1-E6 are depicted in Figure 3, 4,
Sup2, Sup3 and Sup4. The seasonal amplitudes and interannual trends are summarized in
Table 1. The seasonal amplitude of SST (minimum values in February-March around 14-
17 ºC and maximum values in August-September around 20-26ºC) increased eastward
through the S section although the local decrease at S2 (Figure 3 and Sup2, Table 1). The
seasonal changes were larger through the E section (~14 to ~28ºC) and show weaker
spatial variations (Figure 4 and Sup3, Table 1). The SSS (Figure Sup4), do not exhibit a
seasonal cycle well-correlated to the harmonic function Eq. 10 ($r^2 < 0.5$; Table Sup2).
The lower and more spatially stable SSS values were observed along the S section during
the entire period (around 36.0-37.5), while increase with latitude through the E section
(around 36.7-38.1).
The seasonal amplitude of $f\mathrm{CO}_{2,sw}$ (from ~340 to ~460 µatm in the S section and from
~340 to ~470 µatm in the E section) and pH (from ~8.00 to ~8.12 units in the S section
and from ~8.00 to ~7.98 to ~8.13 units in the E section) was strongly linked with those
of SST. It exhibits a west-to-east increment through the S section with the exception at
S2 (Figure 3 and Sup2, Table 1) and remained approximately constant through the E
section (Figure 4 and Sup3, Table 1). These spatial heterogeneities in the seasonal cycles
were found to be leaded by the different rise in SST during late summer along each section
as minimal spatial differences were observed during the rest of the year.





The $C_T$ (Figure Sup4) seasonally decreased from January-February to September-October
(from ~2180 to ~2085 µmol kg$^{-1}$ in the S section and from ~2260 to ~2105 µmol kg$^{-1}$ in
the E section) in phase with the enhancement biological production. The seasonal
amplitude of $C_T$ increased eastward through the S section and northward through the E
section, following the salinification gradient (Figure Sup4, Table 1). Once removed the
effect of salinity, the seasonal cycle of $NC_T$ shows minimal differences in the S section
between the western and the easternmost part, while in the E section the NCT and its
seasonal amplitude continued to northward increase (Figure Sup4, Table 1). The
enhanced adjustment (correlation) of $NC_T$ with Eq. 10 ($0.47<r^2<0.61$ at S section and
$0.70<r^2<0.88$ at E section) compared to $C_T$ ($0.28<r^2<0.56$ at S section and $0.45<r^2<0.73$
at E section) emphasizes the relevance of the salinity-dependent processes. The lower
correlations encountered through the S section shows the higher impact of eventual
processes (i. e. changes in the evaporation/precipitation, river runoff, mesoscale features)
locally modifying the surface carbon system in this area and introducing spatial
heterogeneities in their seasonal cycles.

## 4. Discussion

### 4.1. Spatial characterization of the CO₂ system and its seasonality

The observation-based data allows to evaluate, with high spatio-temporal resolution, the
seasonal cycle of the $CO_2$ system together with their spatial heterogeneities in the Alboran
Sea (S section) and eastern coastal transitional area of the Iberian Peninsula (E section).
The seasonal cycle of the variables considered was subject to spatial variability related to
the irregular coastline of the Iberian Peninsula, which caused local differences in the
oceanographic features and variances in the distance-to-land of the vessel track.
The west-to-east warming and salinification of MAW while entering and advancing
across the Alboran Sea was found to occur mainly during summer and account to rise
eastward the $fCO_{2,sw}$ and fall down the pH (Figure 2). The lowest seasonal amplitude of
$fCO_{2,sw}$ was encountered in the western Alboran Sea (Figure 3). During the late-winter,
the AMI reaches its shallowest position and feed the surface with $CO_2$-rich waters coming
from deeper areas in the Mediterranean Sea (De La Paz et al., 2009; Echevarría et al.,
2002; Gómez-Jakobsen et al., 2019; Minas et al., 1991), elevating $fCO_{2,sw}$ around S1 in
comparison to adjacent waters (Figure 2 and 3). During summer, the wind-induced
upwelling along the northern coast of the western Alboran Sea cooled the surface and

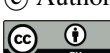



enhanced the biological drawdown of $f$CO$_{2,sw}$ and in C$_T$ (e. g. Bolado-Penagos et al., 2020;
Folkard et al., 1997; Gómez-Jakobsen et al., 2019; Peliz et al., 2009; Richez and
Kergomard, 1990; Stanichny et al., 2005).
The seasonal variability of the AJ (García-Lafuente et al., 2002; Macías et al., 2008, 2016;
Vargas-Yáez et al., 2002) modified the SST signature (Figure 1b) influencing $f$CO$_{2,sw}$ and
pH in the Alboran Sea. The high-intensity of the AJ feeding the two-gyres configuration
during summer (Peliz et al., 2013; Renault et al., 2012) introduced larger spatial changes
compared to the rest of the year. The vessel tracks longitudinally crossed the WAG
through its northern part and followed the northern path of the EAG. The signal of the
summer AJ surrounding the northern part of the WAG (Figure 1b) was observed in local
minimum values of SST and $f$CO$_{2,sw}$ (Figure 2) at S1 (20.68 ± 2.20 ℃ and 401.68 ± 27.13
µatm) and S3 (21.15 ± 2.11 ℃ and 407.30 ± 26.20 µatm), which increased toward the
core of the WAG at S2 (22.63 ± 2.05 ℃ and 429.98 ± 24.86 µatm). The progressively
cooling and decrement in $f$CO$_{2,sw}$ from S2 to S3 (Figure 2) reflects the signal of the cold
and nutrient-rich filament separating the gyres (Gómez-Jakobsen et al., 2019; Millot,

476    1999).

The SST and $f$CO$_{2,sw}$ increased toward the northern path of the EAG around S4 (23.89 ±
2.03 ℃ and 438.25 ± 25.22 µatm) and S5 (24.05 ± 1.61 ℃ and 441.67 ± 16.22 µatm) due
to the mixing of MAW with warmer MW surrounding the Cape of Gata and recirculating
westward along the southern Iberian coastline (Millot, 1999; Sánchez-Garrido et al.,
2013). In terms of C$_T$ and NC$_T$ (Figure 2 and Sup1), a weak decrement around S2 was
observed between January and September and may be due to the injection of deeper
waters into surface waters enhancing the biological drawdown in the core of the WAG.
The C$_T$ and NC$_T$ continue increasing eastward S2 throughout the year as it mixed with
MW.
The hydrodynamic regime of the Alboran Sea during summer with the AJ showing its
maximum intensity (Figure 1b) introduces spatial heterogeneities in the seasonal cycles
(Figure 2). The seasonal amplitudes of SST, $f$CO$_{2,sw}$ and pH (Figure 3 and Sup2, Table 1)
around the WAG (at S2) and EAG (at S4) were higher than the observed over the filament
separating both gyres (at S3). The opposite occurred for C$_T$ (Figure Sup4, Table 1), which
suggests that the upwelled waters transported by the filament were not enough
remineralized to compensate the SST-driven decrease in $f$CO$_{2,sw}$ during summer.



The eastern coastal transitional area of the Iberian Peninsula was subject to variability
related with changes in the intensity, morphology and path of the Northern Current
(Figure 1b). During winter, the warm waters in the wind-shielded area North of Cape of
Nao mixed with cool and salty MW transported by the Northern Current. It explained the
observed decrease in SST of ~1.0ºC during the cold months from Sagunto to Barcelona
coasts (north of S5; Figure 2). During summer, the change in the path of the Northern
Current due to the formation of a thermal front in the axis of the Pyrenees (López-García
et al., 1994) favoured the recent MAW to be northward spreading and to get trapped along
the north-easternmost Iberian coastal area. It forms the warmest waters of the Western
Mediterranean (Lopez Garcia et al., 1994; Millot, 1999) and account to reduce the
observed cooling (~0.8ºC) at this time of the year (Figure 2). In the southernmost part of
the section, the SST increased from Cape of Gata (at S5) to Cape of Nao (at E4) by ~1.5ºC
during summer and decreased by ~0.7ºC during winter (Figure 2). The enhanced
northward spreading of MAW and less wind stress during summer drive the warming,
while a low intense branch of the Northern Current transporting MW and progressing
southward Cape of Nao weakly cool the area during winter (López-García et al., 1994;
López-Jurado et al., 1995).
The offshore recirculation of the Northern Current driven by the bathymetry and the
formation of the high-intense Balearic Front during the warm months (Millot, 1999),
detectable in the reanalysis-based SST map (Figure 1b), explained the local decrease in
SST and $f$CO$_{2,sw}$ observed at E4 (Figure 2). The C$_T$ and NC$_T$ signatures evidenced the
differences between the areas south and north of Cape of Nao (Figure 2 and Sup4). The
northernmost part of the section receives remineralized MW transported by the Northern
Current which elevates C$_T$ and NC$_T$. Ulses et al. (2023) recently suggested that the
convective area in the Gulf of Lion behaves as a source of natural and anthropogenic
carbon to the intermediate waters of the western Mediterranean, which can enter the
surface through vertical mixing and account for the observed high amount of C$_T$ and NC$_T$.
In contrast, the southernmost part was supplied with recent MAW with relatively low C$_T$
and NC$_T$.
The seasonal variations were modulated by the higher stratification during the warm
months and the variety of mesoscale features (mainly meanders and eddies) interacting
with the most energetic Northern Current during the cold months (Bosse et al., 2021;





Millot, 1999). The seasonal amplitudes of SST and $f$CO$_{2,sw}$ increased northward from E1
to E6 (Figure 3 and Sup3, Table 1). The higher seasonal amplitudes occurred in the areas
where the Northern Current introduces larger differences between the cold and warm
months. The location of station E5, away from the influence of the Northern Current
during the warm months, explained its locally lower seasonal amplitudes compared to
adjacent waters. Nevertheless, these heterogeneities were minimal and do not caused
differences in the seasonal amplitude of pH (Table 1). In the case of C$_T$ and NC$_T$ (Figure
Sup4, Table 1), the enhancement in the mixing of MAW with MW during winter
increased northward the seasonality from E1 to E4.
The E6 was subject to local variability related with freshwater discharge from the Ebro
River interacting with the circulation pattern. The Ebro River runoff peaks in March-May
due to the combined action of precipitation during winter and snowmelt in the upper river
basins during spring (Zambrano-Bigiarini et al., 2011). This fed the coastal area around
the Ebro Delta with low SSS and SST waters (see in minimum SST compared to adjacent
waters in February; Figure 1b). The intense NAC at this time of the year further cooled
this coastal area and inflowed saline water which neutralized the peak signal of freshwater
discharge. During summer and fall, the low SSS signal resulted from the Ebro River
runoff combined with the northward spreading of MAW. This explained the minimum
seasonal differences in SSS (Figure Sup4). The approximately constant A$_T$ and NA$_T$
content at E6 throughout the year resulted from the interactions of freshwater fluxes with
MW and MAW compensated for the seasonal variations in C$_T$ and NC$_T$ (Figure Sup4)
expected by air-sea interactions and due to its position over the continental shelf, hence
enhancing biological processes.

**4.2. Warming of the Western Mediterranean Sea and interannual trends of the**

549       **CO$_2$ system variables**

The ongoing warming of the surface Western Mediterranean Basin and its impact on the
marine carbonate dynamics were assessed. The interannual trends are shown in Table 1
and 2. During 2019-2024, the SST increased at a rate of $0.38 \pm 0.05$ ºC yr$^{-1}$ in the S section
and $0.30 \pm 0.04$ ºC yr$^{-1}$ in the E section. The rate of increase in SST locally intensified at
S2 ($0.50 \pm 0.09$ ºC yr$^{-1}$) may be due to the transport and accumulation of surface waters
toward the core of the WAG. Its variability, migration and progressively collapse can also
account for the rapid warming of the area (Sánchez-Garrido et al., 2013; Viúdez et al.,





1998; Vélez-Belchí et al., 2005). Interannual trends were also computed for SST
reanalysis monthly data (0.042° x 0.042°; with dates spanning 24 years within 01/01/2000
and 01/03/2024) from the Med MFC physical multiyear product (Escudier et al., 2020;
2021; Nigam et al., 2021), available at Copernicus Marine Data Store
(https://data.marine.copernicus.eu/products). The SST reanalysis data was interpolated to
the coordinates of the CanOA-VOS data. The SST trends based on CanOA-VOS data
were in the same order of magnitude of those based on reanalysis data for 2019-2024.
Considering the reanalysis data-based SST trends during 2000-2019 in the S section
($0.046 \pm 0.005$ °C yr$^{-1}$, p-value<0.01) and E section ($0.067 \pm 0.005$ °C yr$^{-1}$, p-value<0.01),
the CanOA-VOS data-based SST trends reported a strengthening in warming during
2019-2024 of 87.9% and 78.0% in the respective subregions compared to the previous
two decades. The rates of increase in SST experienced an acceleration of >97% in
comparison with the extracted from the Hadley Centre HadISST1.1 dataset (Rayner et
al., 2003) among the period 1950-2009 for the Atlantic and Mediterranean basin (0.007
°C yr$^{-1}$; p-value < 0.01, and 0.009 °C yr$^{-1}$, p-value > 0.1; respectively; Hoegh-Guldberg et
al., 2014).
The CanOA-VOS data-based interannual SST trends were found to be reinforced during
summer by 55.2% in the S section and by 32.4% in the E section ($0.60 \pm 0.20$ and $0.29 \pm$
$0.10$ °C yr$^{-1}$, respectively; p-values < 0.01) compared to winter ($0.26 \pm 0.04$ and $0.20 \pm$
$0.05$ °C yr$^{-1}$, respectively; p-values < 0.01). The Northern Current cooled the
northernmost part of the E section and accounted to decelerate the warming in comparison
to the S section. These trends enhanced the comprehension of the stronger warming
during the warm season compared to the cold season, as the reanalysis data-based trends
for the same period were not statistically significant (p-values > 0.1). In addition, they
represent an increment in warming of 81-84% respect to 2000-2019 ($0.10 \pm 0.03$ °C yr$^{-1}$,
p-value < 0.05, in the S section; and $0.06 \pm 0.03$ °C yr$^{-1}$, p-value <0.1, in the E section).
Comparisons were difficult to perform during wintertime as non-significant trends were
identified for 2000-2019 (p-values > 0.1). These results emphasized the relevant role of
the large increase in SST during the warm season on the progressing acceleration in
warming. It aligns with projections from climate models for both terrestrial and marine
environments in the mid latitudes, particularly within the Mediterranean region, in
consequence of human-induced global warming, which was detailed by Hoegh-Guldberg
et al., (2018) in the AR6 Synthesis Report (IPCC, 2023). The CanOA-VOS data-based





interannual SST trends reported an increase in SST during the study period of $1.91 \pm 0.26$
ºC in the Alboran Sea and $1.52 \pm 0.22$ ºC along the eastern Iberian coastal transitional
zone. These cumulative increments were 48.3% and 34.94% respectively higher than
those estimated for the global surface ocean from 1850-1900 to 2001-2020 ($0.99 \pm 0.12$
ºC; IPCC, 2023).
The warming contributes to modify the marine carbonate system dynamics, mainly
accelerating the increase in $f\mathrm{CO_{2,sw}}$ and acidification. The interannual trends of $f\mathrm{CO_{2,sw}}$
and pH (Table 1) were more than twice (except for trends at S1) than those reported for
the Northwestern Mediterranean at the DYFAMED site based on the difference between
average data for the periods 1995-1997 and 2013-2015 ($2.30 \pm 0.23$ µatm yr$^{-1}$ and -0.0022
$\pm 0.0002$ units yr$^{-1}$; Merlivat et al., 2018) and for the Northeast Atlantic at the ESTOC
site based on in situ measurements since 1995 ($2.1 \pm 0.1$ µatm yr$^{-1}$ and $0.002 \pm 0.0001$
units yr$^{-1}$, respectively; González-Dávila and Santana-Casiano, 2023). The interannual
rates accelerated eastward along the S section and northward along the E section (Table
1). The stronger trends at S3 compared to adjacent waters (S2 and S4) may be due to the
transport of $\mathrm{CO_2}$-rich waters from the southern Iberian coast through the filament. The
trends in the S section were conducted by the larger rates of change encountered during
the warm season compared to the cold season. The opposite occurred in the E section,
where an intense increase in $f\mathrm{CO_{2,sw}}$ accompanied by a drawdown in pH occurred during
winter and trends were reversed during summer (Table 1).
These spatial differences among the cold and warm seasons were mainly linked with
variations in the biological production/remineralization and mixing and were independent
of the surface ocean warming. Hence, they were required to be assessed together with the
$\mathrm{NC_T}$ trends for a better understanding. The $\mathrm{NC_T}$ interannually decreases throughout the
region (Table 2). The rapid depletion in the S section during winter in comparison to
summer could be due to first, an interannual weakened in remineralization processes
and/or inputs of $\mathrm{CO_2}$-rich water to the area during the cold months, and second, an
interannual strengthened in the biological uptake during the warm months. However,
these variations resulted insufficient to compensate the increase in $f\mathrm{CO_{2,sw}}$ and subsequent
fall down in pH induced by warming during the cold and even more during the warm
months. Conversely, in the E section, the variations in lateral/vertical advection, primary
driven variations in the (sub)mesoscale structures (Alberola et al., 1995; Bosse et al.,



2021; 2016; Bourg and Molcard, 2021), were of high-relevance and introduced
differences in the annual cycle of $NC_T$. The interannual variations during winter (Table
1, Figure Sup4) were minimal likely due to not significant changes in remineralization
and in the dissolved $CO_2$ concentration of waters transported into the area. The decrease
in $NC_T$ intensified during summer (Table 1, Figure Sup4) likely caused by the
enhancement in biological production together with the dismissing lateral advection (this
may be related with a reinforcement in the front formed in the axis of the Pyrenees due
to the increasingly higher SST of the MAW).
Once removed the effects of temperature, the interannual $pH_{19}$ trends overturned to
negligible and were not statistically significant in the S section ($<-0.001$ units yr$^{-1}$; p-
values $> 0.1$). It suggest that warming is directly and indirectly (by rising the $fCO_{2,sw}$)
driving the acidification while the progressively enhancing in biological productivity
compensates for the expected fall down in pH driven by rising atmospheric $CO_2$. In the E
section, $pH_{19}$ were reduced by 63% ($-0.002 \pm 0.001$ units yr$^{-1}$; p-values $< 0.01$) in
comparison to the pH trends, which explains that the increase in SST is contributing more
than half on the acidification due to only the atmospheric $fCO_2$ increase. The negative
$pH_{19}$ trends reinforced in the E section by 47% during the cold season due to the
enhancement in remineralization. The $pH_{19}$ trends reversed to positive during the warm
season due to the important role of biological production actively reducing $fCO_{2,sw}$ and
rising pH at this time of the year. This remarked the relevant role of non-thermal processes
occurring during the cold season and contributing to the acidification trends on an
interannual scale (see below)
However, despite the high statistical confidence in the trends and the consistency found
with reanalysis products, the acceleration in surface warming and consequent changes in
$fCO_{2,sw}$ and pH observed may be linked to isolated extreme events such as marine heat
waves and are not necessarily indicative of prolonged behaviours over time. The globally
increased frequency and magnitude in marine heat waves in phase with warming (Oliver
et al., 2018; Hoegh-Guldberg et al., 2018; Frölicher et al., 2018; Smale et al., 2019) could
feedback and hence continue expediting the ocean warming. The influence of these
extreme events is especially relevant in semi-enclosed seas as the Mediterranean,
recognized as one of the most affected marine areas as yearly mentioned in the



Copernicus Ocean State Reports (OSR; EU Copernicus Marine Service;
https://marine.copernicus.eu/access-data/ocean-state-report) since 2016 (OSR1-OSR7).

### 4.3. The relative contribution of thermal and non-thermal processes on the surface $f\text{CO}_{2,\text{sw}}$

The relative influence of thermal and non-thermal processes on the $f\text{CO}_{2,\text{sw}}$ variations at
seasonal and interannual scales were addressed following the procedures of Takahashi et
al. (2002) and Fassbender et al. (2022), hereinafter referred as T'02 and F'22,
respectively. Its temporal evolution is depicted in Figures 3 and 4 and show the high
coincidence between both methodologies. The average $f\text{CO}_{2,\text{sw}}$ explained by thermal and
non-thermal processes ($f\text{CO}_{2,\text{T}}$ and $f\text{CO}_{2,\text{NT}}$, respectively) presented differences lower
than 5 µatm between T'02 and F'22 (Table 2). The consistency with the widely employed
T'02 engenders confidence in the validity and reliability of the most updated F'22
method.
The seasonal amplitudes and interannual trends of $f\text{CO}_{2,\text{T}}$ and $f\text{CO}_{2,\text{NT}}$ are presented in
Table 2. The thermal-driven seasonal changes ($\text{d}f\text{CO}_{2,\text{T}}$) were found to approximately
double those independent of temperature ($\text{d}f\text{CO}_{2,\text{NT}}$) throughout the region. The seasonal
variations were close to twice in the E section compared to the S section. The T/B ratios
(Table 2) demonstrated the control of thermal processes over the seasonality of $f\text{CO}_{2,\text{sw}}$
throughout the region. The T/B ratios in the westernmost part of the S section (between
1 and 2) were consistent with previous studies in the Strait of Gibraltar (Curbelo-
Hernández et al., 2021; De La Paz et al., 2009). The T/B ratios increased eastward as the
AJ advanced in the Alboran Sea and caused by the intense increase in $\text{d}f\text{CO}_{2,\text{T}}$ compared
to $\text{d}f\text{CO}_{2,\text{NT}}$. They exceeded 2 in S4-S5 and E1-E6, which demonstrated the larger control
of SST over $f\text{CO}_{2,\text{sw}}$ in areas less influenced by incoming of surface Atlantic water
The interannual trends show the control of thermal processes over the increase in $f\text{CO}_{2,\text{sw}}$
during 2019-2024 (Figure 3 and 4; Table 2). The strong and statistically significant
interannual $f\text{CO}_{2,\text{T}}$ trends show the important role of warming in elevating $f\text{CO}_{2,\text{sw}}$. The
weak and non-significant $f\text{CO}_{2,\text{NT}}$ trends suggest that spatio-temporal variations in the
biological processes, circulations patterns and air-sea gas exchange introduced local
differences in the distribution of $f\text{CO}_{2,\text{sw}}$. It difficult to assess the impact of the non-
thermal processes on an interannual scale at each of the stations. The interannual trends



of $f$CO$_{2,T}$ and $f$CO$_{2,NT}$ for the entire S and E sections (Table 2) were statistically significant
at more than the 95% level of confidence and its coupling described, with less than 0.3
µatm yr$^{-1}$ of difference (<1%), the interannual rates of $f$CO$_{2,sw}$ during 2019-2024 (Table
1; section 4.2).
The thermal processes govern the changes in $f$CO$_{2,sw}$ on an interannual scale with a
contribution ranged between ~76-92% in the S section and ~73-83% in the E section. The
contributions for $f$CO$_{2,NT}$ were between ~8-25% and ~17-27%, respectively. The decrease
in $f$CO$_{2,NT}$ compensated by ~6-30% the increase in $f$CO$_{2,sw}$ at S1-S5 and E1-E2, while its
increase contributed by ~24-53% to rise $f$CO$_{2,sw}$ at E3-E6. The negative $f$CO$_{2,NT}$ trends in
the S section were related to progressive enhancement in the biological uptake (mainly
during spring/summer) not compensated by remineralization and/or vertical/lateral
advections of remineralized waters (mainly during autumn/winter) in areas influenced by
recent MAW. Conversely, the interannual increase in $f$CO$_{2,NT}$ in the E section suggest that
the supply of cool and remineralized MW along the path of the high-intense Northern
Current surpasses the biological drawdown of surface CO$_2$ and is accounting to accelerate
the increase in $f$CO$_{2,sw}$ on an interannual scale.
**4.4. Mechanism controlling the seasonality of $f$CO$_{2,sw}$**
The partial contribution of the individual component controlling the seasonal cycle of
$f$CO$_{2,sw}$ was assessed. The seasonal rates of change of $f$CO$_{2,sw}$ ($\frac{\mathrm{d}f\mathrm{CO}_{2,sw}}{\mathrm{dt}}$, hereinafter
d$f$CO$_2$) explained by fluctuations in SST ($\frac{\partial f\mathrm{CO}_{2,sw}}{\partial \mathrm{SST}}\frac{\partial \mathrm{SST}}{\mathrm{dt}}$, hereinafter d$f$CO$_2^{\mathrm{SST}}$), SSS
($\frac{\partial f\mathrm{CO}_{2,sw}}{\partial \mathrm{SSS}}\frac{\partial \mathrm{SSS}}{\mathrm{dt}}$, hereinafter d$f$CO$_2^{\mathrm{SSS}}$), A$_T$ ($\frac{\partial f\mathrm{CO}_{2,sw}}{\partial \mathrm{A}_T}\frac{\partial \mathrm{A}_T}{\mathrm{dt}}$, hereinafter d$f$CO$_2^{\mathrm{AT}}$) and C$_T$
($\frac{\partial f\mathrm{CO}_{2,sw}}{\partial \mathrm{C}_T}\frac{\partial \mathrm{C}_T}{\mathrm{dt}}$, hereinafter d$f$CO$_2^{\mathrm{CT}}$) were calculated for each year using Eq. 7 (section
2.3.3) at S1-S5 and E1-E6 and depicted in Figure 5. The positive values indicate an
increase in $f$CO$_{2,sw}$ from February to September, while negative values the opposite.
The SST was identified as the main driver of d$f$CO$_2$, describing 45-78% and 55-83% of
its changes in the S and E sections, respectively. In the S section (Figure 5a), d$f$CO$_2^{\mathrm{SST}}$
increased westward as MAW get warmed in the Alboran Sea, while the incursion of the
filament locally cooled the surface and decreased d$f$CO$_2^{\mathrm{SST}}$ at S3. In the E section (Figure
5b), d$f$CO$_2^{\mathrm{SST}}$ increased northward and reach its maximum north of Cape of Nao (at E4-



E6), particularly during 2021-2022 (32.0-32.5 µatm month$^{-1}$), due the higher influence of
warmed MW.
The $A_T$ has a low influence on increasing d$f$CO$_2$ in the entire region (<15%). As the
$f$CO$_{2,sw}$ inversely changes with $A_T$, the weakly negative d$f$CO$_2^{AT}$ found for some years
along the S section show fluctuations in the periods of increment and decrement of $A_T$
likely related with changes in the mixing processes. The $A_T$ contribution becomes
negligible at E6 (<1%) due to the minimal seasonal amplitude of $A_T$ and $NA_T$ (Figure
Sup4). The approximately constant $A_T$ and $NA_T$ levels throughout the year may be due to
the bicarbonate and carbonate content from the Ebro River runoff being neutralized by
those in MW and MAW, which spread into the area during winter and summer,
respectively. d$f$CO$_2^{AT}$ tend to decrease since 2020-2021 in S1-S3, S5 and E1 due to the
progressively weakening in the $NA_T$ depletion from February to September. The opposite
occurred north of Cape of Palos, where the seasonal cycle of $NA_T$ reaches its maximum
amplitude (20-27 µmol kg$^{-1}$ at E3 and E4). The interannual dealkalinization in S and E
sections (Table 1) behaves as a source of heterogeneities: the interannual negative $NA_T$
trends during the cold months (p-values < 0.01) were stronger than during the warm
months (p-values > 0.1) and consistent in both sections. The spatial differences in the
summer trends (weaker in the S compared to E section) account for an enhanced reduction
of the seasonal amplitude of $NA_T$ in the S section.
The d$f$CO$_2^{SSS}$ were minimal in both the S and E sections (<0.7 and < 1.9 µatm month$^{-1}$,
respectively) and show the weak impact of SSS over d$f$CO$_2$ (<3.5%). The entrance of
MAW and its mixing with saltier MW in the Alboran Sea do not allow to identify a
seasonal pattern in SSS (Figure Sup4), thus explained the negligible contribution of SSS
in the S section (~2.3% at S1 which fall down to <1.0% at S2-S5). The larger seasonal
amplitudes of SSS at E1-E5 (Figure Sup4) led a relatively major influence of SSS (~1.0-
2.4% during most of the years). The low seasonal amplitude of SSS and $A_T$ at E6, likely
related with an approximately constant influence of the Northern Current at this location
throughout the annual cycle, caused a minimal variation in d$f$CO$_2$ (<1%).
The depletion in $C_T$, mainly drove by the increased biological production from February
to September, had a significant impact on d$f$CO$_2$ (25-38%). It compensates more than one
third of the expected increase in d$f$CO$_2$ driven by SST and slightly prompt by $A_T$. In the
S section (Figure 5a), the lower changes observed during the period of study in d$f$CO$_2^{CT}$



(4-6 µatm month$^{-1}$) compared to d$f$CO$_2$$^{SST}$ (6-9 µatm month$^{-1}$) demonstrated that
fluctuations in C$_T$ were increasingly insufficient to counterbalance the warming-driven
increase in d$f$CO$_2$, even at S2-S4 where the biological production enhanced and hence the
d$f$CO$_2$$^{CT}$ reinforced since 2020. In the westernmost part of the S section, the influence of
C$_T$ offsetting d$f$CO$_2$ was maximum during 2019-2020 at S1 (>84%), S2 (67.3%) and S3
(86.1%) and diminished toward 2023 (37.1%, 38.3% and 45.1%, respectively). In the
easternmost part, this compensation was around 33-44% at S4-S5 throughout the period
(as at S2 and S3 since 2020) except for 2023 at S5, in which d$f$CO$_2$$^{CT}$ weakened and offset
only the 22.8%. In the E section (Figure 5b), the progressively strength in the processes
depleting C$_T$ throughout the period at E1-E4 and since 2020 at E5-E6 compensated by
33-46% the d$f$CO$_2$$^{SST}$, which changes inversely to d$f$CO$_2$$^{CT}$. The lowest compensation
found in 2019 at E5 (28.8%) and E6 (18.4%) was likely related with isolated eventual
improved injections of remineralized waters along the Northern Current path, which
offset the biological uptake of C$_T$ and elevated the d$f$CO$_2$$^{CT}$.
**4.5. Air-sea CO$_2$ exchange across the Western Boundary of the Mediterranean**

760        **Sea**

The Eastern Boundary of the Mediterranean Sea was characterized for the first time in
terms of air-sea CO$_2$ exchange. The variability of FCO$_2$ was governed by fluctuations in
$\Delta f$CO$_2$ (Figure 6), mainly controlled by the larger range of variation of $f$CO$_{2,sw}$ (325-500
µatm) compared to $f$CO$_{2,atm}$ (390-425 µatm). The SST fluctuations has a relevant role by
primary controlling $f$CO$_{2,sw}$ (section 4.3) and modulating the solubility of CO$_2$ at the air-
sea interface, while the changes in the wind speed influence the gas transfer velocity
(Wanninkhof, 2014).
The entire monitored area was undersaturated for CO$_2$ respect to the low atmosphere
between late October and June ($\Delta f$CO$_2$= -35.30 ± 8.97 µatm), acting as an atmospheric
CO$_2$ sink (-2.56 ± 0.55 mmol m$^{-2}$ d$^{-1}$) which peaks in winter (-4.53 ± 0.44 and -3.29 ±
0.31 mmol m$^{-2}$ d$^{-1}$ in S and E sections, respectively). During summer, the area was
supersaturated for CO$_2$ ($\Delta f$CO$_2$= 36.43 ± 0.35 µatm) and acted as a source, which was
about three times more intense along the E section (1.70 ± 0.43 mmol m$^{-2}$ d$^{-1}$) compared
to the S section (0.57 ± 0.35 mmol m$^{-2}$ d$^{-1}$). The spatial differences in SST during warm
months introduced heterogeneities in the seasonal outgassing among both sections: the
higher SST during summer in the E section reduced the solubility and contributed to a



higher increase in $f\mathrm{CO}_{2,sw}$ respect to $f\mathrm{CO}_{2,atm}$ ($\Delta f\mathrm{CO}_2$= 49.83 ± 0.32 µatm) compared to
the cooler S sectiom ($\Delta f\mathrm{CO}_2$= 16.35 ± 0.14 µatm). The seasonality in the formation of the
$\mathrm{CO}_2$ sink and source in the Alboran Sea was consistent with previous studies in the Strait
of Gibraltar (Curbelo-Hernández et al., 2021; de la Paz et al., 2011, 2009) and Northwest
African coastal transitional area in the Northeast Atlantic (Curbelo-Hernández et al.,
2021b; Padin et al., 2010) and agreed with the seasonal pattern characteristic for tropical
and subtropical regions (Bates et al., 2014; Takahashi et al., 2002). The warming during
summer at S1 was insufficient to led supersaturated conditions ($\Delta f\mathrm{CO}_2$= -5.56 ± 0.26
µatm) and thus acted as a $\mathrm{CO}_2$ sink throughout the year (-2.83 ± 1.77 mmol m$^{-2}$ d$^{-1}$ during
cold months and -0.52 ± 0.02 mmol m$^{-2}$ d$^{-1}$ during the warm months), which coincided
with the behaviour observed in the Strait of Gibraltar during 2019 (Curbelo-Hernández et
al., 2021). The sink and source status during cold and warm months encountered in the
Eastern Iberian Margin agreed with $\mathrm{FCO}_2$ evaluations based on observations in the
Mediterranean basin through its northwestern (Wimart-Rousseau et al., 2023, 2021, 2020)
and eastern parts (Sisma-Ventura et al., 2017), and confirms previous estimations based
on satellite data and models (D'Ortenzio et al., 2008; Taillandier et al., 2012).
The variations in $\mathrm{FCO}_2$ during the period of study were addressed by averaging the data
across seasons and years at each of the selected stations (Figure 7). The same procedure
was applied to $\Delta f\mathrm{CO}_2$ and wind speed (Figure Sup5 and Sup6). The evolution of the
seasonal ingassing and outgassing was evaluated by computing interannual trends for
average $\mathrm{FCO}_2$ and $\Delta f\mathrm{CO}_2$ (Figure 7). The interannual $\mathrm{FCO}_2$ trends evidenced the
progressively strength of the summer source in the S section, which was accelerated at
S2 in response to the enhanced warming around the WAG (detailed in section 4.2) and at
S4-E1 due to their exposition to increasing wind forcing (Figure Sup5 and Sup6). It was
caused by the increase in $f\mathrm{CO}_{2,sw}$ during the warm months not offset by biological
drawdown which elevated $\Delta f\mathrm{CO}_2$. In contrary, the localization of E2-E6 over the eastern
Iberian continental shelf and slope allowed the relevant biological uptake at this time of
the year to compensate for the influx of $\mathrm{CO}_2$-rich water. It introduced heterogeneities in
$\Delta f\mathrm{CO}_2$ between years which do not allow to identify statistically significant trends.
During spring and autumn, the increase in $\Delta f\mathrm{CO}_2$, mainly driven by warming,
accompanied by the decreasing wind stress (Figure Sup5 and Sup6), led the positive
interannual $\mathrm{FCO}_2$ trends at S2-S5 and E1-E6 (Figure 7). They show the weakening in the



ingassing during autumn and the achievement of a near-equilibrium state with the atmosphere during spring by the end of the study period. The $FCO_2$ reversed to weakly positive during spring 2023 in the E section, which prolonged the seasonal source period having a relevant impact on the net annual $FCO_2$. During winter, the increasing wind forcing compensated the reduction in the ingassing expected by the rise in $\Delta fCO_2$ (Figure Sup5 and Sup6). However, the variability in the wind speed and other processes involved in the non-thermal change of $fCO_{2,sw}$ between years does not allowed the identification of statistically significant rates of change in the $CO_2$ sink status. Particularly, the relatively high wind speed during winter 2021 may have contributed to accelerated horizontal transports, increasing $fCO_{2,sw}$ and hence $\Delta fCO_2$ (Figure Sup5 and Sup6).

The predominantly negative $FCO_2$ during most of the year led a net annual $CO_2$ sink behaviour. The positive $FCO_2$ trends during summer, spring and autumn have forced the annual average $CO_2$ invasion to decrease by 44-65% at S2-S5 (ranging from $-0.66 \pm 0.06$ and $-0.84 \pm 0.04$ mol m$^{-2}$ during 2019 to $-0.27 \pm 0.09$ and $-0.47 \pm 0.09$ mol m$^{-2}$ during 2023) and by 60-80% at E1-E6 (ranging from $-0.32 \pm 0.09$ and $-0.53 \pm 0.09$ mol m$^{-2}$ during 2019 to $-0.11 \pm 0.10$ and $-0.13 \pm 0.09$ mol m$^{-2}$ during 2023). The unique hydrodynamic of the Strait of Gibraltar strongly influenced the air-sea $CO_2$ exchange at S1: the ingassing during summer partially compensated for the reduction of the annual influx and resulted in a lower increase in $FCO_2$ (23%) from 2019 ($-0.77 \pm 0.02$ mol m$^{-2}$ yr$^{-1}$) to 2023 ($-0.60 \pm 0.06$ mol m$^{-2}$ yr$^{-1}$).

Considering the annual average $FCO_2$ for the S and E section, the net ingassing have decreased at a rate of $0.11 \pm 0.02$ mol m$^{-2}$ yr$^{-1}$ yr$^{-1}$ (p-value<0.01) in the Alboran Sea and by $0.08 \pm 0.02$ mol m$^{-2}$ yr$^{-1}$ yr$^{-1}$ (p-value<0.01) in the Eastern Iberian Margin. It contrast with the strength of the $CO_2$ sink across the western Mediterranean basin recently reported by Zarghamipour et al., (2024) for 1984-2019 based on a combination of observational data and model simulations ($0.007 \pm 0.001$ mol m$^{-2}$ yr$^{-1}$ yr$^{-1}$). Additionally, Zarghamipour et al., (2024) noted the reduction of the annual net $CO_2$ source behaviour of the Central Mediterranean basin at an estimated rate of $0.003 \pm 0.001$ mol m$^{-2}$ yr$^{-1}$. The findings suggest that the acceleration in the increase in $fCO_{2,sw}$ induced by the rapid warming, together with the progressive reduction in solubility, is reversing the interannual $FCO_2$ trends compared to previous decades, may be causing the study area to be resemble the Central and Eastern Mediterranean basin in terms of air-sea $CO_2$ exchange. The reduction



of the net annual invasion was consistent with previous estimations in such coastal and
shelf environments across the eastern tropical and subtropical South Atlantic during
2002-2018 (between $0.03 \pm 0.01$ and $0.09 \pm 0.02$ mol m$^{-2}$ yr$^{-1}$ yr$^{-1}$; Ford et al., 2022) and
toward mid-latitudes over the Scotian Shelf (with average FCO$_2$ ranging from -1.7 mol
m$^{-2}$ yr$^{-1}$ yr$^{-1}$ in 2002 to -0.02 mol m$^{-2}$ yr$^{-1}$ yr$^{-1}$ in 2006; Sisma-Ventura et al., 2017). The
continuation of this decreasing rate for net annual ingassing would imply the reversion of
the study area to a net annual CO$_2$ source behaviour before 2030.
The net CO$_2$ invasion was calculated by integrating the annual cycle of FCO$_2$ during
2019-2023. The net FCO$_2$ in the Alboran Sea was $-1.57 \pm 0.49$ mol m$^{-2}$ yr$^{-1}$, which
represented a strength in the CO$_2$ sink in comparison with adjacent surface areas across
the Strait of Gibraltar (between -0.82 and -1.01 mol m$^{-2}$ yr$^{-1}$ during 2019-2021; Curbelo-
Hernández et al., 2021) and the Eastern Iberian Upwelling (-1.33 mol m$^{-2}$ yr$^{-1}$; Chen et
al., 2013). The net FCO$_2$ along the Eastern Iberian margin was $-0.70 \pm 0.54$ mol m$^{-2}$ yr$^{-1}$,
which fall within the range of those modelled for the deep-convection area around the
Bay of Marseille (Northwestern Mediterranean Basin) during 2012-2013 (-0.5 mol m$^{-2}$
yr$^{-1}$; Ulses et al., 2023) and estimated based on observations during 2017-2018 (between
-0.26 and -0.81 mol m$^{-2}$ yr$^{-1}$; Wimart-Rousseau et al., 2020). However, it was opposite to
the net outgassing across the Easten Mediterranean basin ($0.85 \pm 0.27$ mol m$^{-2}$ yr$^{-1}$ during
2009-2015; Sisma-Ventura et al., 2017). The net CO$_2$ sink for the monitored area across
the Alboran Sea (14,000 Km$^2$) and eastern Iberian margin (40,000 Km$^2$) was $-0.97 \pm 0.30$
Tg CO$_2$ yr$^{-1}$ ($-0.26 \pm 0.08$ Tg C yr$^{-1}$) and $-1.22 \pm 0.95$ Tg CO$_2$ yr$^{-1}$ ($-0.33 \pm 0.25$ Tg C yr$^{-1}$
$^{1}$). These findings powerfully contributed to the assessment of the air-sea CO$_2$ exchange
in the Mediterranean basin (Borges et al., 2005) and global coastal and shelf areas (Chen
et al., 2013).
**5. Conclusion**
The five years of automatically underway observations through the CanOA-VOS line
provided a high spatio-temporal resolution dataset which includes the surface physical
and MCS properties across the western boundary of the Mediterranean Sea. It allowed
the characterization, with an improved degree of certainty for the highly variable Alboran
Sea and Eastern Iberian coastal transitional area, of patterns and mechanisms involved on
seasonal and interannual timescales.





The findings reveal the influence of the upper-layer circulation patterns and subsequent
physical and biological implications on the MCS. In the Alboran Sea, the high intensity
of the AJ during summer warms the surface layer, driving larger seasonal changes in SST,
$f\mathrm{CO_{2,sw}}$ and pH toward the core of the WAG and EAG. Meanwhile, the intensified
filaments cool the surface at this time of the year and reduce these seasonal amplitudes in
the area between both gyres. The seasonality of the Northern Current meridionally
separates the eastern Iberian coastal transitional area at Cape of Nao: the northernmost
part, fed with cool, salty and remineralized MW during the cold season and influenced
by the northward spreading of MAW during the warm season, show the largest seasonal
amplitudes for SST, $f\mathrm{CO_{2,sw}}$, pH and $\mathrm{C_T}$ compared to the southernmost part, supplied with
recent MAW during most of the year and by a weak and relatively warmed branch of the
Northern Current during winter.
Even with the limitations of five-year observational period, the interannual trends report
the relevant acceleration in warming in comparison with the previous two decades (78-
88%). The SST increased at rates ranging between 0.26 and 0.43 ºC yr$^{-1}$ and drove a rapid
increase in $f\mathrm{CO_{2,sw}}$ within 4.18 and 5.53 µatm yr$^{-1}$ and a decrease in pH within -0.0049
and -0.0065 units yr$^{-1}$. The strengthening of interannual variations during the study period
was primarily conducted by the reinforcement of trends, within one-third to one-half,
during the warm season in comparison to the cold season. The $\mathrm{NC_T}$ decreased at a rate
between -0.5 and -1.6 µmol kg$^{-1}$, suggesting an interannual dismiss in the
remineralization/biological production ratio. These progressively variations were
counterbalanced along the Eastern Iberian margin by the increasingly relevance of
lateral/vertical advection and mesoscale structures, which favours the inflow of
remineralized waters mainly during the cold season.
The variations in $f\mathrm{CO_{2,sw}}$ were found to be strongly controlled by temperature
fluctuations. On a seasonal scale, the rapidly warmed AJ as enters the Alboran Sea drives
a significant eastward increase in $\mathrm{d}f\mathrm{CO_{2,T}}$ compared to $\mathrm{d}f\mathrm{CO_{2,NT}}$. Consequently, the
thermal-driven seasonal changes intensified and doubled those non-thermal as MAW
formed, advanced northward along the eastern Iberian margin and mixed with MW. The
driver analysis has identified the SST as the primary driver of the seasonality of $f\mathrm{CO_{2,sw}}$,
accounting for 45-83% of its variations. The processes controlling the $\mathrm{C_T}$ offsets 25-38%
of the seasonal amplitude of $f\mathrm{CO_{2,sw}}$ expected by the effect of thermal-processes. The
seasonal variations in $\mathrm{A_T}$ infers minor changes in $f\mathrm{CO_{2,sw}}$ (<15%) while the contribution





of SSS fluctuations was close to negligible (<3.5%). The seasonal amplitude of $f$CO$_{2,sw}$
increased during the study period in the Alboran Sea, while high mesoscale variability
along the Eastern Iberian margin infers higher ranges of uncertainties and do not allow to
obtain relevant conclusions. Based on the driver analysis, this variation was driven, in
first term, by the increasing contribution of temperature (due to the seasonal amplitude of
SST is becoming larger) and, in second term, by the decreasing contribution of $C_T$ (due
to the dismissing remineralization/production ratio). On an interannual scale, the ~76-
92% of the increase in $f$CO$_{2,sw}$ was described by warming. In the Alboran Sea and
extending northward to Cape of Palos, non-thermal processes, primarily biological
drawdown during spring blooms, compensated for up to one-third of the expected
increase in $f$CO$_{2,sw}$ due to rising SST. The opposite occurred north of Cape of Palos, where
non-thermal processes, mainly the inflow of CO$_2$-rich MW during the cold season,
accounted for the increase in $f$CO$_{2,sw}$.
The assessment of the air-sea CO$_2$ exchange shows the Western boundary of the
Mediterranean basin undersaturated and acting as a significant sink for atmospheric CO$_2$
during most of the year, while presented supersaturated conditions which led a CO$_2$
source status during the warm months. On an annual basis, the entire monitored area acted
as a net CO$_2$ sink. The evolution of the FCO$_2$ has shown a reduction in the net annual CO$_2$
invasion at statistically significant rates ranging between 0.06 and 0.13 mol m$^{-2}$ yr$^{-1}$ yr$^{-1}$
(40-80% since 2019), which would reverse the behaviour of the area to a net annual CO$_2$
source before 2030 if the climate conditions continues the nowadays trends. The
weakening in the net annual CO$_2$ sink was driven by the ongoing strength of the summer
outgassing (mainly in the Alboran Sea) and the weakening in the autumn and spring
ingassing (throughout the region). Integrating the annual cycle of FCO$_2$ during the entire
study period, net CO$_2$ ingassing calculated for the Alboran Sea and Eastern Iberian
Margin was -1.57 ± 0.49 and -0.70 ± 0.54 mol m$^{-2}$ yr$^{-1}$.
The present investigation has addressed the need to design and implement systematic
observation strategies for characterizing the physico-chemical seawater properties in the
Mediterranean basin, an effort that has been required by the scientific community for the
last decades. This research pretended to emphasize the efficiency of VOS in the
monitoring of the surface physical and MCS variables, particularly in areas subject to
high variability under anthropogenic pressure as coastal regions and semi-enclosed seas,
where the implementation of other observation-based alternatives is challenging. The



results improve the comprehension of the MSC dynamics along a coastal transitional area
in the Western Mediterranean Sea, which is of high environmental and socio-economic
importance and significantly influences the European climate. Likewise, they contribute
to a more accurate understanding of the role of coastal areas in the context of Global
Change at both basin and global scales. Although the study period was relatively short
and larger time-series are necessary for quantifying long-term trends and making future
projections, it has encompassed drastic variations compared to previous decades likely
caused by isolated events feedbacked by climate change (i. e. marine heat waves). This
has enabled the study of physicochemical dynamics under conditions expected for the
future state of the ocean.
**Code Availability**
The        $CO_{2,SYS}$        programme        for        MATLAB        is        available        at
https://github.com/jonathansharp/CO2-System-Extd.
**Data Availability Statement**
The underway observations provided by the SOOP CanOA-VOS in the Western
Mediterranean Sea (February 2019 – February 2024) used in this investigation are
published in open-access at Zenodo (doi.org/10.5281/zenodo.13379011) and available in
since September 2023 at the ICOS Data Portal (https://www.icos-cp.eu/data-
products/ocean-release). The SST reanalysis monthly data (0.042º x 0.042º) from the Med
MFC physical multiyear product (Escudier et al., 2020; 2021; Nigam et al., 2021) are
available at Copernicus Marine Data Store (https://data.marine.copernicus.eu/products).
ERA5 hourly wind speed reanalysis data at 10 m above the sea level used to calculate air-
sea    $CO_2$    fluxes    is    available    at    Copernicus    Climate    Data    Store
(https://cds.climate.copernicus.eu/).
**Author contribution**
All the authors made significant contributions on this research.  M. G.-D., J. M. S.-C. and
A.G.G. installed and maintained the equipment in the VOS. D. C-H and D. G-S participated
in routine maintenance and data acquisition. D. C.-H. developed the MATLAB® routines
and conducted the data processing and analysis. All authors contributed to the writing of
the manuscript and supported its submission.



**Declaration Competing interest**

The authors declare that the research was conducted in the absence of any commercial or financial relationships that could be construed as a potential conflict of interest.

**Acknowledgement**

This research was supported by the Canary Islands Government and the Loro Parque Foundation through the CanBIO project, CanOA subproject (2019–2024), and the CARBOCAN agreement (Consejería de Transición Ecológica y Energía, Gobierno de Canarias). We would like to thank the JONA SOPHIE ship owner, Reederei Stefan Patjens GmbH & Co. KG, the NISA-Marítima company and the captains and crew members for the support during this collaboration. Special thanks to the technician Adrian Castro-Álamo for biweekly equipment maintenance and discrete sampling of total alkalinity aboard the ship. The SOOP CanOA-VOS line is part of the Spanish contribution to the Integrated Carbon Observation System (ICOS-ERIC; https://www.icos-cp.eu/) since 2021 and has been recognized as an ICOS Class 1 Ocean Station. The participation of D. C-H was funded by the PhD grant PIFULPGC-2020-2 ARTHUM-2



**Legend for Figures**

Figure 1. (a) Map of the Western boundary of the Mediterranean Sea with the CanOA-VOS tracks between February 2019 and February 2024 (red) and the location of the stations of interest along the southern (S1-S5) and eastern (E1-E6) sections. The main Capes and Gulf along the geographically rugged Iberian coastline are shown. The schematic diagram summarized the classical circulation patterns: in the Alboran Sea (blue), the Atlantic Jet (AJ) surrounds the Western and Eastern Anticyclonic Gyres (WAG and EAG, respectively) and forms Modified Atlantic Water (MAW), while along the Eastern Iberian margin (purple), the Mediterranean Water (MW) is transported from the Northwestern Mediterranean basin along the path of the Northern Current. The northward spreading of MAW during summer and southward spreading MW during winter is depicted with dashed arrows. The thermal front formed in the axis of the Pyrenees during summer is depicted with a black dashed line. (b) SST maps built with reanalysis monthly data (0.042° x 0.042°) for February and September 2023 from the Med MFC physical multiyear product (Escudier et al., 2020; 2021; Nigam et al., 2021), available at Copernicus Marine Data Store (https://data.marine.copernicus.eu/products).

Figure 2. Spatial distribution of the average SST, $f\mathrm{CO_{2,sw}}$, pH, and $C_T$ calculated on a seasonal and annual basis every 0.1° longitude along the S section (left panels) and every 0.25° latitude along the E section (right panels). The 3-months periods January-March, April-June, July-September and October-December were considered as winter, spring, summer and autumn, respectively. Note the different scales used for $C_T$ due to significant variations between the S and E sections. Standard deviations are provided in Table Sup1 and indicate the range of variability among the study period.

Figure 3. Time-series of SST, $f\mathrm{CO_{2,sw}}$ and pH at S1, S3 and S5 along the eastern Iberian margin within the five years of observations. The weekly average data was fitted to harmonic Eq. 10. The thermal and non-thermal terms of the average $f\mathrm{CO_{2,sw}}$ calculated by following the procedures of Takahashi et al., 2002 (T,02) and Fassbender et al., 2022 (F'22) and the $\mathrm{pH_{19}}$ are depicted. The coefficients $a$-$f$, standard errors of estimate and $r^2$ given by Eq. 10 are presented in Table Sup1.

Figure 4. Time-series of SST, $f\mathrm{CO_{2,sw}}$ and pH at E1, E4 and E5 in the Alboran Sea within the five years of observations. The weekly average data was fitted to harmonic Eq. 10. The thermal and non-thermal terms of the average $f\mathrm{CO_{2,sw}}$ calculated by following the



procedures of Takahashi et al., 2002 (T,02) and Fassbender et al., 2022 (F'22) and the
pH$_{19}$ are depicted. The coefficients *a-f*, standard errors of estimate and r$^2$ given by Eq. 10
are presented in Table Sup1.
Figure 5. Temporal evolution of the seasonal rates of $f$CO$_{2,sw}$ explained by each of its
drivers within the five years of observation. The differences between monthly average
data for February and September (where minimum and maximum SST and $f$CO$_{2,sw}$ were
encountered) was considered to compute the seasonal trends. The standard deviation of
the monthly average data were considered in the calculation of the seasonal changes and
infers errors in the computation of $f$CO$_{2,sw}$, which are summarized in Table Sup3. The
cumulative $f$CO$_{2,sw}$ change resulting from the distinct impulsors $\frac{\mathrm{d}f\mathrm{CO}_{2,sw}}{\mathrm{d}t}$ (sum) were
consistent with the observed seasonal $\Delta f$CO$_2$ trends ($\frac{\mathrm{d}f\mathrm{CO}_{2,sw}}{\mathrm{d}t}$ (obs)), thereby instilling
confidence in the methodology.
Figure 6. Temporal variations of CO$_2$f (blue; left axis), $\Delta f$CO$_2$ (orange; right axis) and
wind speed (gray; left axis) at (a) S1-S5 and (b) E1-E6. A piecewise polynomial-based
smoothing spline was applied to the weekly average data (represented with dots). Gaps
were covered by the harmonic fitting (Eq. 10; dash line). The black lines represent the
interannual increase in CO$_2$f. The seasonally-detrended interannual rates of change of
CO$_2$f and $\Delta f$CO$_2$ are shown in each panel. *** denotes that the trends are statistically
significant at the 99% level of confidence, ** at the 95% level of confidence and * at the
90% level of confidence. The wind speed does not show statistically significant
interannual trends (p-values > 0.1).
Figure 7. Temporal evolution of average CO$_2$f calculated on a seasonal and annual basis
for each year (2019-2023) at S1-S5 and E1-E6. Same representation for $\Delta f$CO$_2$ and wind
speed is available in Figure Sup5 and Sup6. The 3-months periods January-March, April-
June, July-September and October-December were considered as winter, spring, summer
and autumn, respectively. The legend includes the interannual trends for CO$_2$f (mol m$^{-2}$
yr$^{-1}$) based on linear regression of the seasonal and annual means. *** denotes that the
trends are statistically significant at the 99% level of confidence, ** at the 95% level of
confidence and * at the 90% level of confidence. Standard deviations are presented in
Table Sup4.
**Legend for Tables**



Table 1. Seasonal amplitudes and interannual trends of SST, SSS, $f\mathrm{CO}_{2,sw}$, pH, $\mathrm{pH}_{19}$, $\mathrm{C_T}$
and $\mathrm{NC_T}$. The seasonal changes were calculated as the amplitude of Eq. 10 fitted to the
weekly average data at each station. The error of the seasonal amplitudes was assumed as
the product of the standard error of estimate given by the harmonic function by 2. The
interannual changes were based on linear regressions and given for each station and for
the entire S and E sections (considering the total amount of average data at S1-S5 and E1-
E6, respectively) during the cold and warm season. The interannual trends of SST during
2000-2019 (based on reanalysis monthly data from the Med MFC physical multiyear
product [Escudier et al., 2020; 2021; Nigam et al., 2021]; detailed in section 4.2) was
included for comparison. The trends were obtained by the linear regressions of the
seasonally-detrended weekly average data and include their standard error of estimate.
*** denotes that the trends are statistically significant at the 99% level of confidence, **
at the 95% level of confidence and * at the 90% level of confidence.
Table 2. Means, seasonal amplitudes and interannual rates of change of thermal and non-
thermal components of $f\mathrm{CO}_{2,sw}$ ($f\mathrm{CO}_{2,T}$ and $f\mathrm{CO}_{2,NT}$, respectively) calculated by following
Takahashi et al., 2002 and Fassbender et al., 2022 (T'02 and T'22, respectively). The
seasonal changes were calculated as the amplitude of Eq. 10 fitted to the weekly average
data at each station. The error of the seasonal amplitudes was assumed as twice the
standard error of estimate given by the harmonic function. The trends were obtained by
the linear regressions of the seasonally-detrended weekly average data and include their
standard error of estimate. *** denotes that the trends are statistically significant at the
99% level of confidence, ** at the 95% level of confidence and * at the 90% level of
confidence.



Fig. 1





Fig. 2





Fig. 3

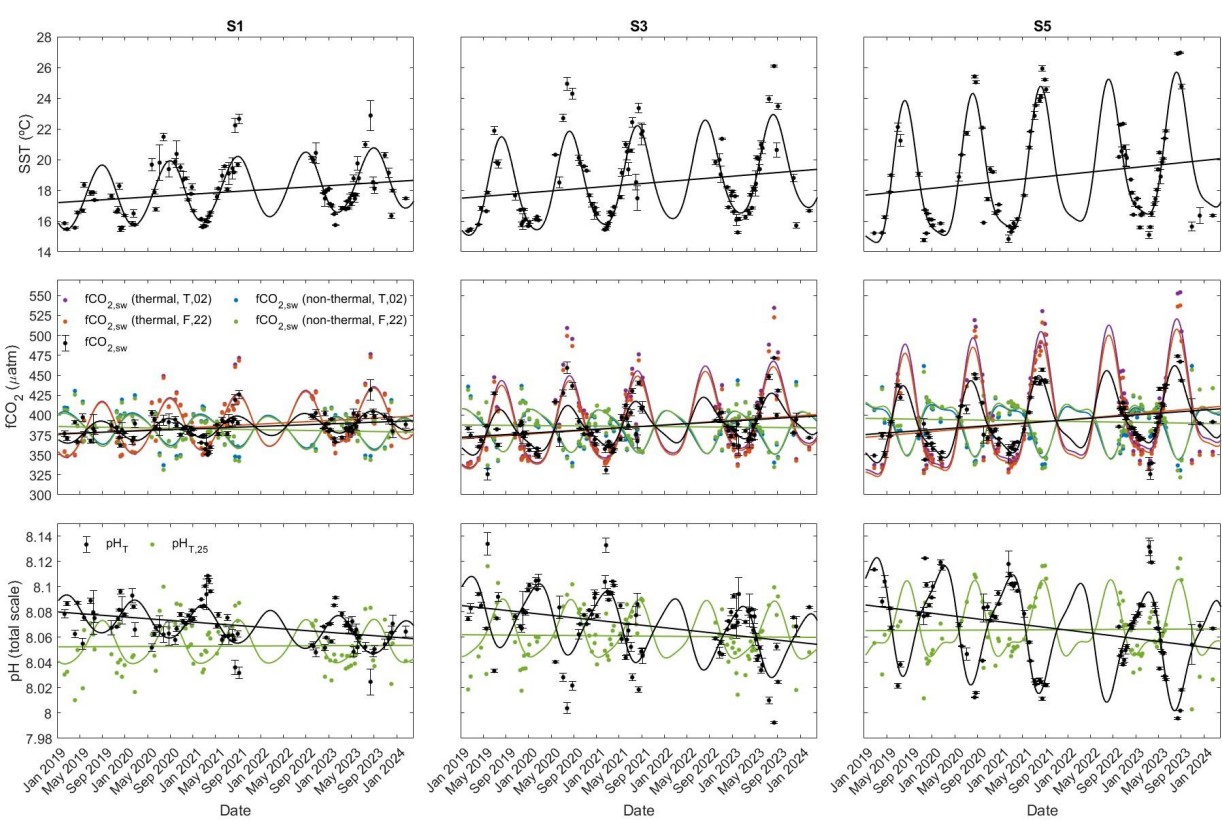





Fig. 4

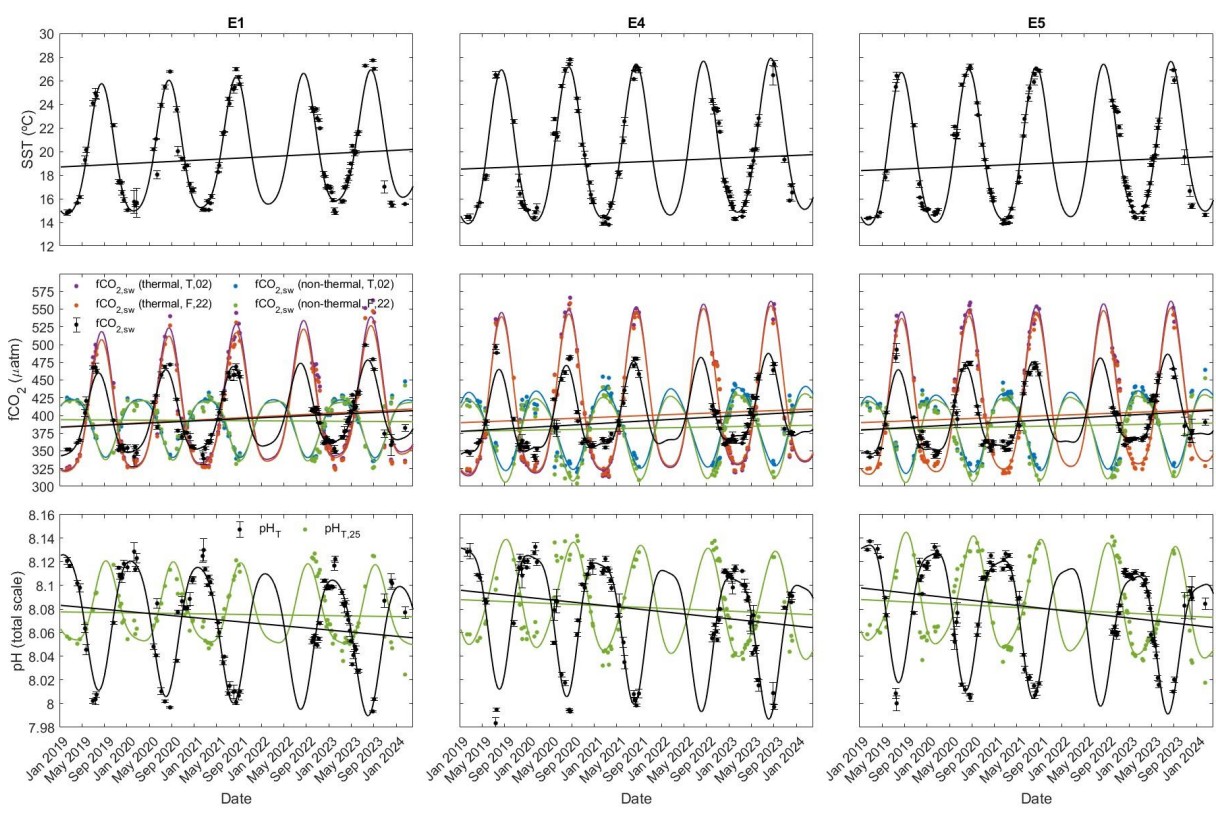



Fig. 5

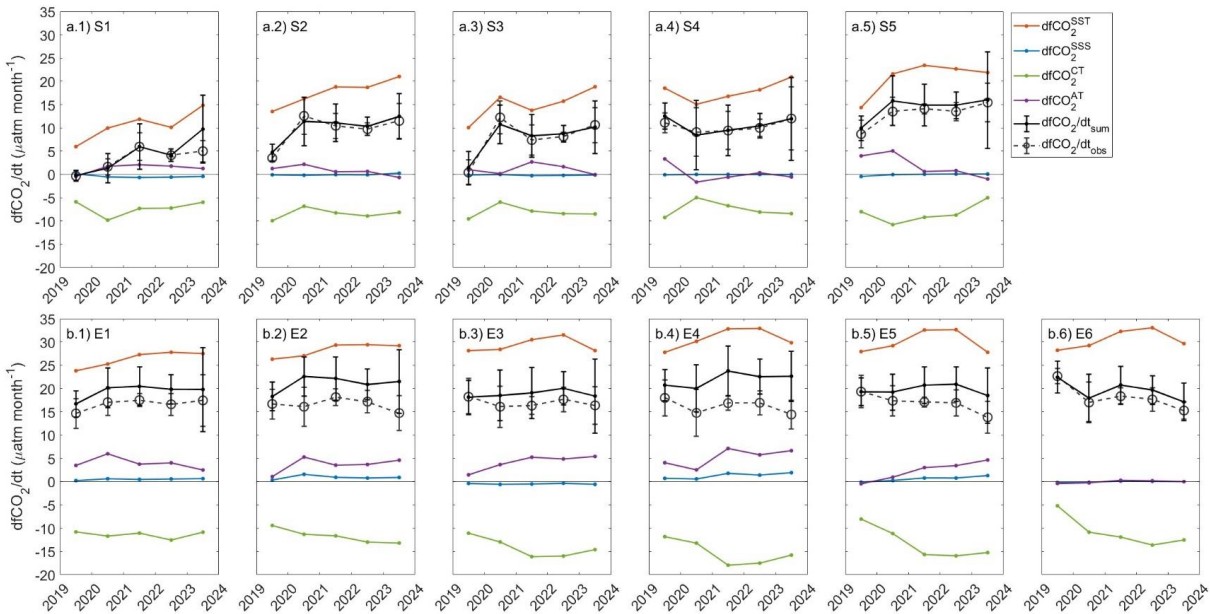



Fig. 6

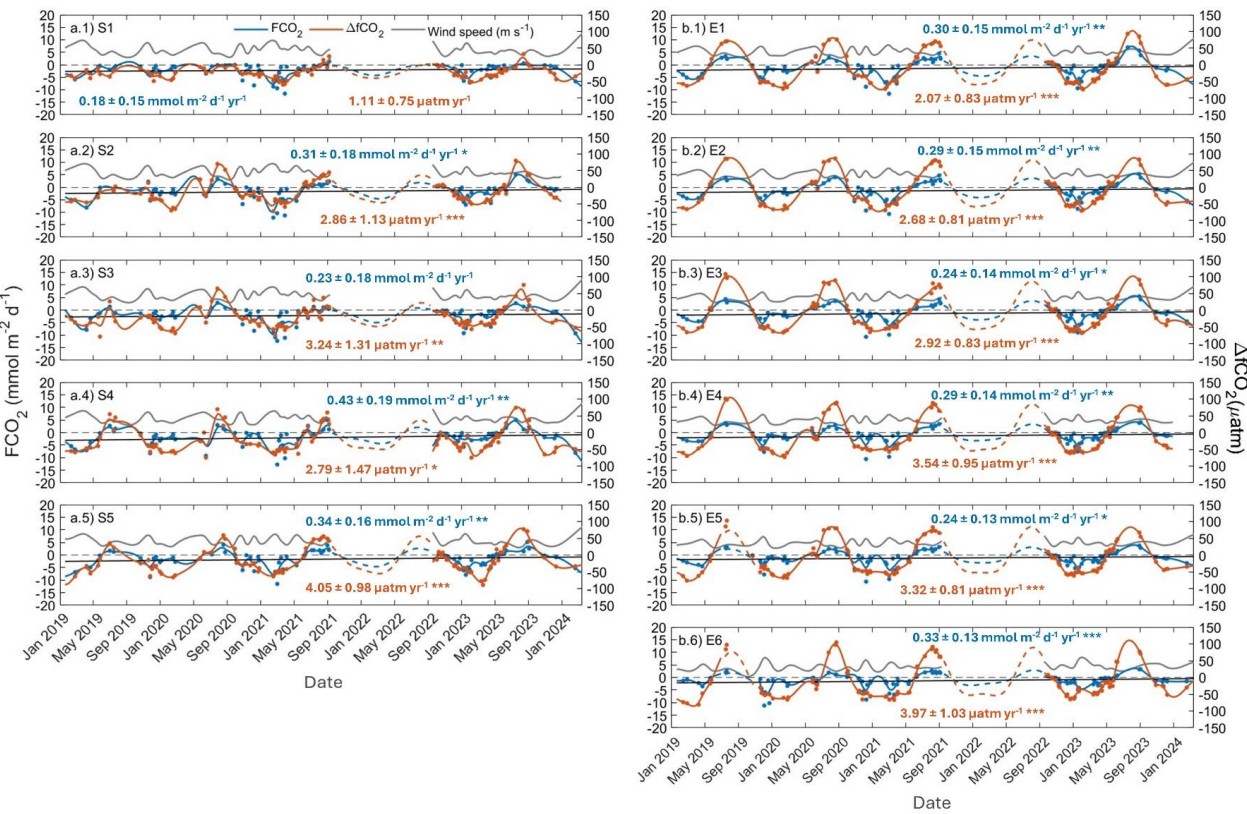





Fig. 7

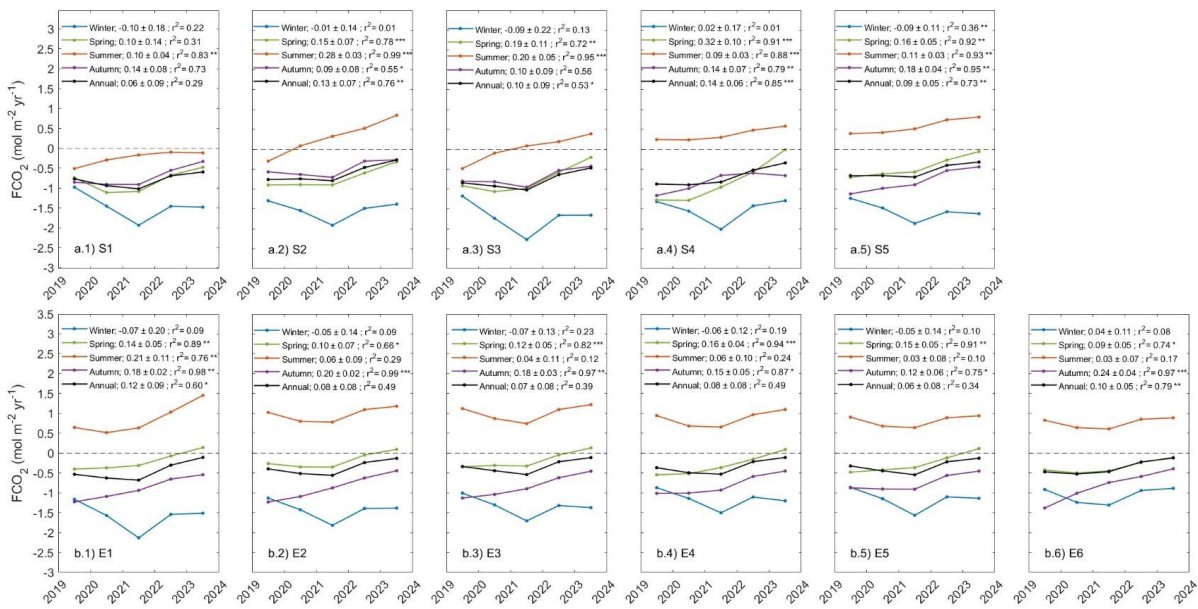





Table 1

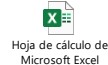

Hoja de cálculo de Microsoft Excel

| | SST | | SSS | | $f\mathrm{CO_{2sw}}$ | | pH | | pH$_{is}$ | | C$_t$ | | NC$_t$ | | A$_t$ | | NA$_t$ | |
|---|---|---|---|---|---|---|---|---|---|---|---|---|---|---|---|---|---|---|
| | Seasonal amplitude (°C) | ratio (°C yr⁻¹) | Seasonal amplitude | ratio (yr⁻¹) | Seasonal amplitude (µatm) | ratio (µatm yr⁻¹) | Seasonal amplitude (total scale) | ratio (units yr⁻¹) | Seasonal amplitude (total scale) | ratio (units yr⁻¹) | Seasonal amplitude (µmol kg⁻¹) | ratio (µmol kg⁻¹ yr⁻¹) | Seasonal amplitude (µmol kg⁻¹) | ratio (µmol kg⁻¹ yr⁻¹) | Seasonal amplitude (µmol kg⁻¹) | ratio (µmol kg⁻¹ yr⁻¹) | Seasonal amplitude (µmol kg⁻¹) | ratio (µmol kg⁻¹ yr⁻¹) |
| S1 | 4.21 ± 1.90 | 0.28 ± 0.07 *** | 0.293 ± 0.328 | -0.074 ± 0.012 *** | 27.78 ± 20.27 | 3.13 ± 0.75 *** | 0.0300 ± 0.0210 | -0.0040 ± 0.0008 *** | 0.0344 ± 0.0280 | 0.0002 ± 0.0010 | 41.2 ± 16.3 | -6.4 ± 1.0 *** | 26.8 ± 16.3 | -2.2 ± 0.6 *** | 29.4 ± 32.8 | -7.4 ± 1.2 *** | 10.6 ± 5.9 | -2.7 ± 0.4 *** |
| S2 | 7.50 ± 2.18 | 0.50 ± 0.09 *** | 0.158 ± 0.258 | -0.078 ± 0.010 *** | 70.20 ± 28.27 | 4.68 ± 1.10 *** | 0.0674 ± 0.0254 | -0.0055 ± 0.0010 *** | 0.0582 ± 0.0292 | 0.0022 ± 0.0011 * | 37.3 ± 19.4 | -7.9 ± 1.1 *** | 35.4 ± 19.4 | -3.5 ± 0.8 *** | 15.6 ± 25.9 | -7.9 ± 1.0 *** | 5.6 ± 4.7 | -2.9 ± 0.4 *** |
| S3 | 6.42 ± 2.38 | 0.36 ± 0.09 *** | 0.333 ± 0.334 | -0.070 ± 0.012 *** | 57.23 ± 35.36 | 5.12 ± 1.32 *** | 0.0563 ± 0.0340 | -0.0059 ± 0.0013 *** | 0.0455 ± 0.0276 | -0.0004 ± 0.0010 | 47.4 ± 17.6 | -5.7 ± 1.1 *** | 33.0 ± 17.6 | -1.8 ± 0.7 *** | 33.4 ± 33.7 | -7.0 ± 1.3 *** | 12.1 ± 6.1 | -2.6 ± 0.5 *** |
| S4 | 7.53 ± 2.58 | 0.26 ± 0.10 *** | 0.344 ± 0.457 | -0.051 ± 0.017 *** | 74.89 ± 38.91 | 4.89 ± 1.45 *** | 0.0698 ± 0.0372 | -0.0053 ± 0.0014 *** | 0.0544 ± 0.0242 | -0.0014 ± 0.0009 | 43.0 ± 19.9 | -3.6 ± 1.6 *** | 33.0 ± 19.9 | -0.6 ± 0.7 | 34.7 ± 46.3 | -5.2 ± 1.7 *** | 12.5 ± 8.2 | -1.9 ± 0.6 *** |
| S5 | 9.25 ± 2.34 | 0.45 ± 0.09 *** | 0.562 ± 0.575 | -0.062 ± 0.022 *** | 96.99 ± 25.18 | 6.17 ± 0.98 *** | 0.0940 ± 0.0242 | -0.0067 ± 0.0009 *** | 0.0601 ± 0.0304 | 0.0003 ± 0.0012 | 50.8 ± 24.3 | -5.6 ± 2.0 *** | 34.3 ± 24.3 | -2.0 ± 0.9 *** | 56.6 ± 58.0 | -6.3 ± 2.2 *** | 20.1 ± 10.3 | -2.3 ± 0.8 *** |
| summer | | 0.59 ± 0.20 *** | | | | 7.23 ± 2.33 *** | | -0.0069 ± 0.0020 *** | | 0.0020 ± 0.0014 | | -3.9 ± 1.4 *** | | -2.1 ± 0.7 *** | | -3.1 ± 2.2 | | -1.1 ± 0.8 |
| winter | | 0.26 ± 0.04 *** | | -0.094 ± 0.020 *** | | 3.43 ± 0.96 *** | | -0.0047 ± 0.0011 *** | | -0.0006 ± 0.0010 | | -7.8 ± 1.8 *** | | -2.4 ± 0.8 *** | | -9.5 ± 2.0 *** | | -3.4 ± 0.7 *** |
| total | | 0.38 ± 0.05 *** | | -0.065 ± 0.009 *** | | 4.76 ± 0.59 *** | | -0.0054 ± 0.0006 *** | | 0.0002 ± 0.0005 | | -5.7 ± 0.8 *** | | -2.0 ± 0.4 *** | | -6.6 ± 0.9 *** | | -2.4 ± 0.3 *** |
| 2000-2019 | | 0.03 ± 0.00 *** | | | | | | | | | | | | | | | | | |
| E1 | 11.07 ± 2.15 | 0.28 ± 0.08 *** | 0.522 ± 0.463 | -0.069 ± 0.017 *** | 116.94 ± 23.18 | 4.44 ± 0.85 *** | 0.1148 ± 0.0234 | -0.0052 ± 0.0009 *** | 0.0670 ± 0.0206 | -0.0008 ± 0.0008 | 81.1 ± 15.1 | -5.5 ± 1.4 *** | 52.6 ± 15.1 | -1.5 ± 0.6 *** | 52.7 ± 47.0 | -7.0 ± 1.7 *** | 18.3 ± 8.2 | -2.4 ± 0.6 *** |
| E2 | 11.64 ± 1.82 | 0.31 ± 0.07 *** | 0.482 ± 0.486 | -0.094 ± 0.018 *** | 121.57 ± 21.54 | 4.79 ± 0.81 *** | 0.1172 ± 0.0218 | -0.0059 ± 0.0008 *** | 0.0732 ± 0.0190 | -0.0011 ± 0.0007 | 83.2 ± 13.5 | -7.4 ± 1.5 *** | 56.8 ± 13.5 | -1.9 ± 0.5 *** | 48.8 ± 49.2 | -9.5 ± 1.9 *** | 16.9 ± 8.5 | -3.3 ± 0.6 *** |
| E3 | 12.44 ± 1.89 | 0.24 ± 0.07 *** | 0.592 ± 0.604 | -0.138 ± 0.023 *** | 124.78 ± 21.85 | 4.99 ± 0.82 *** | 0.1225 ± 0.0204 | -0.0067 ± 0.0008 *** | 0.0818 ± 0.0236 | -0.0031 ± 0.0009 *** | 94.1 ± 21.4 | -10.2 ± 2.0 *** | 63.9 ± 21.4 | -2.0 ± 0.8 *** | 60.0 ± 61.2 | -14.0 ± 2.3 *** | 20.6 ± 10.5 | -4.8 ± 0.8 *** |
| E4 | 13.04 ± 1.80 | 0.23 ± 0.07 *** | 0.768 ± 0.493 | -0.068 ± 0.018 *** | 120.73 ± 25.43 | 5.40 ± 0.94 *** | 0.1196 ± 0.0234 | -0.0061 ± 0.0009 *** | 0.0891 ± 0.0280 | -0.0024 ± 0.0010 *** | 120.1 ± 21.6 | -4.4 ± 1.7 *** | 75.1 ± 21.6 | -0.4 ± 0.8 | 77.9 ± 49.9 | -6.9 ± 1.8 *** | 26.5 ± 8.5 | -2.3 ± 0.6 *** |
| E5 | 12.92 ± 1.74 | 0.23 ± 0.06 *** | 0.538 ± 0.467 | -0.097 ± 0.017 *** | 118.88 ± 21.72 | 5.31 ± 0.79 *** | 0.1165 ± 0.0194 | -0.0064 ± 0.0007 *** | 0.0914 ± 0.0270 | -0.0029 ± 0.0010 *** | 98.4 ± 20.8 | -6.6 ± 1.6 *** | 69.3 ± 20.8 | -0.9 ± 0.7 | 54.6 ± 47.3 | -9.9 ± 1.7 *** | 18.5 ± 8.0 | -3.3 ± 0.6 *** |
| E6 | 13.13 ± 2.02 | 0.19 ± 0.07 *** | 0.108 ± 0.551 | -0.011 ± 0.015 | 124.68 ± 30.17 | 6.09 ± 0.99 *** | 0.1159 ± 0.0256 | -0.0061 ± 0.0008 *** | 0.0929 ± 0.0328 | -0.0032 ± 0.0011 *** | 63.3 ± 27.4 | 0.9 ± 1.6 | 59.3 ± 27.4 | 1.6 ± 0.9 | 10.0 ± 54.7 | -1.2 ± 1.4 | 3.4 ± 9.2 | -0.4 ± 0.5 |
| summer | | 0.29 ± 0.09 *** | | -0.069 ± 0.042 * | | -2.30 ± 1.02 ** | | 0.0011 ± 0.0008 | | 0.0037 ± 0.0012 *** | | -8.5 ± 3.2 *** | | -4.3 ± 0.9 *** | | -7.0 ± 4.3 | | -2.4 ± 1.5 |
| winter | | 0.20 ± 0.04 *** | | -0.092 ± 0.023 *** | | 5.44 ± 0.41 *** | | -0.0067 ± 0.0005 *** | | -0.0036 ± 0.0007 *** | | -5.8 ± 2.1 *** | | -0.4 ± 0.8 | | -9.4 ± 2.4 *** | | -3.2 ± 0.8 *** |
| total | | 0.30 ± 0.04 *** | | -0.082 ± 0.013 *** | | 5.16 ± 0.37 *** | | -0.0061 ± 0.0004 *** | | -0.0022 ± 0.0004 *** | | -5.8 ± 1.1 *** | | -0.9 ± 0.4 *** | | -8.4 ± 1.3 *** | | -2.9 ± 0.4 *** |
| 2000-2019 | | 0.05 ± 0.01 *** | | | | | | | | | | | | | | | | | |





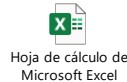

Hoja de cálculo de Microsoft Excel

Table 2

| | $fCO_{2,sw}$ (thermal) | | | | | | $fCO_{2,sw}$ (non-thermal) | | | | | | T/B ratio | |
| | T02 | | | F22 | | | T02 | | | F22 | | | | |
| | Mean (µatm) | Seasonal Amplitude (µatm) | Interannual ratio (µatm yr⁻¹) | Mean (µatm) | Seasonal Amplitude (µatm) | Interannual ratio (µatm yr⁻¹) | Mean (µatm) | Seasonal Amplitude (µatm) | Interannual ratio (µatm yr⁻¹) | Mean (µatm) | Seasonal Amplitude (µatm) | Interannual ratio (µatm yr⁻¹) | T02 | F22 |
|---|---|---|---|---|---|---|---|---|---|---|---|---|---|---|
| S1 | | 70.35 ± 16.39 | 4.53 ± 1.21 *** | | 68.40 ± 15.85 | 4.42 ± 1.17 *** | | 41.04 ± 14.67 | -1.53 ± 1.08 | | 41.44 ± 14.70 | -1.35 ± 1.09 | 1.71 | 1.65 |
| S2 | | 129.76 ± 19.66 | 8.50 ± 1.53 *** | | 124.45 ± 18.69 | 8.18 ± 1.46 *** | | 66.85 ± 17.44 | -3.83 ± 1.36 *** | | 67.53 ± 15.90 | -3.50 ± 1.24 *** | 1.94 | 1.84 |
| S3 | 392.04 ± 40.87 | 109.35 ± 21.50 | 6.04 ± 1.60 *** | 389.02 ± 39.15 | 104.93 ± 20.41 | 5.80 ± 1.52 *** | 386.13 ± 18.44 | 54.11 ± 15.14 | -0.79 ± 1.13 | 386.62 ± 18.77 | 54.02 ± 14.31 | -0.68 ± 1.06 | 2.02 | 1.94 |
| S4 | | 131.09 ± 23.60 | 4.36 ± 1.76 ** | | 125.63 ± 22.41 | 4.19 ± 1.67 *** | | 59.99 ± 14.14 | 0.82 ± 1.05 | | 61.82 ± 13.68 | 0.74 ± 1.02 | 2.19 | 2.03 |
| S5 | | 163.37 ± 20.70 | 7.79 ± 1.61 *** | | 154.95 ± 19.60 | 7.43 ± 1.52 *** | | 65.16 ± 17.94 | -1.31 ± 1.39 | | 68.85 ± 16.76 | -1.25 ± 1.30 | 2.51 | 2.25 |
| summer | | | 11.83 ± 3.68 *** | | | 11.09 ± 3.39 *** | | | -2.92 ± 1.23 ** | | | -3.94 ± 1.44 *** | | |
| winter | | | 3.97 ± 0.70 *** | | | 3.81 ± 0.74 *** | | | -0.62 ± 1.16 | | | -0.29 ± 1.03 | | |
| total | | | 6.20 ± 0.81 *** | | | 5.94 ± 0.77 *** | | | -1.27 ± 0.57 ** | | | -1.14 ± 0.55 ** | | |
| E1 | | 196.07 ± 19.09 | 5.11 ± 1.40 *** | | 186.41 ± 18.12 | 4.84 ± 1.32 *** | | 81.74 ± 10.92 | -0.18 ± 0.80 | | 83.96 ± 10.83 | -0.53 ± 0.79 | 2.40 | 2.22 |
| E2 | | 206.32 ± 16.29 | 5.29 ± 1.23 *** | | 196.92 ± 15.51 | 5.09 ± 1.17 *** | | 89.84 ± 9.73 | -0.07 ± 0.73 | | 91.97 ± 9.61 | -0.32 ± 0.72 | 2.30 | 2.14 |
| E3 | 400.22 ± 70.68 | 219.12 ± 16.15 | 3.86 ± 1.20 *** | 399.02 ± 67.76 | 213.60 ± 15.79 | 3.80 ± 1.17 *** | 389.61 ± 32.15 | 99.59 ± 13.95 | 1.43 ± 1.03 | 385.00 ± 33.52 | 105.81 ± 13.10 | 1.21 ± 0.97 | 2.20 | 2.02 |
| E4 | | 230.66 ± 15.37 | 3.75 ± 1.13 *** | | 222.49 ± 14.90 | 3.68 ± 1.10 *** | | 110.60 ± 15.39 | 1.58 ± 1.13 | | 116.16 ± 15.56 | 1.67 ± 1.15 | 2.09 | 1.92 |
| E5 | | 229.35 ± 14.52 | 3.64 ± 1.05 *** | | 219.99 ± 14.06 | 3.55 ± 1.01 *** | | 108.60 ± 15.35 | 1.93 ± 1.11 * | | 115.03 ± 14.74 | 1.72 ± 1.06 | 2.11 | 1.91 |
| E6 | | 231.16 ± 17.30 | 2.88 ± 1.28 ** | | 221.64 ± 16.61 | 2.84 ± 1.23 ** | | 104.92 ± 19.24 | 3.37 ± 1.33 *** | | 109.10 ± 18.90 | 3.25 ± 1.29 *** | 2.20 | 2.03 |
| summer | | | 6.41 2.13 *** | | | 5.85 2.07 *** | | | -4.79 1.12 *** | | | -6.62 1.63 *** | | |
| winter | | | 2.78 0.60 *** | | | 2.77 0.59 *** | | | 2.91 0.83 *** | | | 2.62 0.67 *** | | |
| total | | | 4.10 0.52 *** | | | 3.95 0.51 *** | | | 1.33 0.46 *** | | | 1.19 0.47 *** | | |



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
