# Peer review of "Spatio-temporal variations in surface Marine Carbonate"

_EGUsphere, 2024_

## Author Response (AR1)

**Author's response to the reviews**

We prodive in this document the point-by-point response (in blue) to the comments by the two anonymous reviewers. The revisions discussed herein have already been incorporated into the updated version of the manuscript. Substantial changes in structure and narrative have been made in accordance with suggestions by both reviewers. These changes have resulted in a more coherent, focused, and accessible manuscript, while maintaining the scientific integrity and depth of the study. The main structural changes implemented in the revised manuscript are summarized below:

- Part of the descriptive content about the study region originally included in the Introduction and Discussion has been moved to a new subsection in the Methodology titled *Study Area* (Section 2.1 in the updated version).

- The former section *Data Adjustments and Statistical Procedures* (Section 2.3.1 in the previous version) has been moved to *Appendix A* to streamline the main text. Additionally, we have included *Appendix B*, which provides a detailed description of the uncertainty estimation for the air–sea $CO_2$ fluxes associated with uncertainties in $fCO_{2,sw}$ and $fCO_{2,atm}$, according to suggestions by reviewer 2.

- Several structural aspects of the *Methods* and *Results* sections have been improved, including the merging of the previous two results subsections into a single Section 3.

- The structure and narrative of the *Discussion* have been substantially revised. Subsections 4.1.1 and 4.1.2 have been introduced under Section 4.1 to better organize the discussion of the results according to the two distinct study regions (Alboran Sea and eastern Iberian margin).

- The *Conclusions* section has been shortened, restructured, and clarified to provide a more concise and effective closing to the manuscript.

Several important methodological aspects, including the calculation of $A_T$, the multivariable Taylor expansion, as well as certain statistical and error propagation calculations, have also been revised and modified in accordance with the reviews.

**Point-by-point response to Reviewer 1**

This study is an important example of analysis of oceanic carbonate systems based on observation data from the western Mediterranean Sea, and the measurement methods and data processing are generally appropriate. However, the structure of the Discussion and Conclusion is poor, and it is very difficult to understand the novelty that should be claimed in this paper. The authors should significantly revise the structure of the Discussion and Conclusion to clarify the appeal points of this paper to the readers.

Thank you very much for your thoughtful and constructive feedback on our manuscript. We greatly appreciate your recognition of the significance of our observation-based analysis of the ocean carbonate system in the Mediterranean Sea. We have carefully considered your comments and suggestions, which have contributed to enhancing the quality and reliability of the manuscript. Below, we provide a point-by-point response to each of your comments.

**Major comments**

Most description in the Discussion and Conclusions are repetitions of the Result. For example, Chapters 4.1 and 4.2 are unnecessary and should be deleted. The current description that merely lists data is redundant, and make it difficult to understand new findings that should be claimed in the paper. The structure of these sections should be substantially revised by deleting unnecessary descriptions.

We have thoroughly revised and modified these sections. The descriptive part has been relocated to a new subsection in the methodology titled "Study Area," and we have aimed to be more concise in discussing the main results presented in Sections 4.1 and 4.2. Additionally, we have modified Section 3: the results are now presented more succinctly, as they will be analysed in depth in the discussion section. In the conclusion, we have removed certain numerical results that contributed to noise and could confuse the reader, as well as some redundant phrases previously addressed in earlier sections. After implementing these changes, the new version of the manuscript is much clearer and can be better understood by the reader.

The multivariable Taylor expansion is performed in Equation (6), but in this paper, TA is calculated as a linear equation of salinity, so there should be very strong multicollinearity between SSS and TA. Therefore, I am very suspicious of the results of this equation. We have to clear the problem of multicollinearity by removing one of the variables or by using methods to avoid multicollinearity (e.g., PLS regression).

We agree that calculating alkalinity from salinity is a relevant point of this paper and significantly influences subsequent calculations, requiring special attention and detailed explanation. Due to limitations in autonomously collecting simultaneous seawater samples for $A_T$ and $C_T$ determination alongside $x CO_2$ measurements, we manually collected discrete samples at different times of the year along the vessel track (102 discrete samples in total with in situ SST and SSS measurements taken during February 2020, March 2021, and October 2023; see Section 2.3 for details). With these empiric data, we obtained a statistically significant linear AT-SSS relationship (Eq. 1) at the 99% confidence level with a high correlation ($r^2 = 0.92$), which we used to calculate alkalinity at the time, latitude, and longitude of the surface xCO2, SST, and SSS observations. We

have previously applied this procedure in the Northeastern Atlantic (Curbelo-Hernández et al., 2021a) and the Strait of Gibraltar (Curbelo-Hernández et al., 2021b), both regions also monitored by CanOA-VOS. However, in those cases, AT fitted better with SSS using a second-degree polynomial equation, consistent with relationships reported for the Atlantic by Lee et al., 2006. In contrast to Lee et al. (2006), a multiparametric regression incorporating SST as a second variable (see Table 1 in Lee et al., 2006) did not yield satisfactory results in our case. After extensive analysis, we determined that our AT observations are correlated the best only with SSS. We attribute this outcome to the fact that our measurements were taken in coastal transition zones, where SST is highly variable and extends beyond the SST ranges established by Lee et al. (2006), who developed their relationships using observations in open-ocean Atlantic areas with more stable temperatures (see Zones 1 and 3 in Figure 1 of Lee et al., 2006).

Although reconstructing $A_T$ through its relationship with SSS has been widely applied to calculate other variables of the Marine Carbonate System and to derive conclusions on oceanic $CO_2$ and pH levels and trends (e.g., Takahashi et al., 2014), we recognize that this method has certain limitations. Primarily, it do not consider biological processes that alter $A_T$ and cannot be traced by salinity (Wolf-Gladrow et al., 2007) as well as the input of dissolved carbonate minerals and bicarbonate-carbonate species from river runoff, sediments and water mixing. Consequently, $A_T$-SSS relationships provide a useful general approximation in regions with stable conditions and less influenced by these processes but carry uncertainties in areas subject to variability.

In our study area along the western boundary of the Mediterranean Sea, surface $A_T$ dynamics is primarily governed by the influx of fresher, low-$A_T$ Atlantic waters and the significant role of evaporation/precipitation (Cossarini et al., 2015). In contrast, terrestrial and riverine contributions of $A_T$ to the Alboran Sea and Eastern Iberian coast were found to have minimal influence on $A_T$ distribution compared to marginal and coastal areas in the eastern Mediterranean Basin (see Table 2 and Figure 1 in Cossarini et al., 2015). Additionally, in the eastern Mediterranean, limited nutrient inflow and reduced water renewal amplify the role of biological processes in regulating carbon and alkalinity cycles. Conversely, in the western Mediterranean, the effect of these biological cycles on alkalinity is relatively diminished due to the influx of cooler, nutrient-rich Atlantic waters, which reduces the relative importance of local biological contributions to alkalinity dynamics. These processes explain the pronounced west-to-east surface gradient in $A_T$ across the Mediterranean basin (Cossarini et al., 2015) and the relatively homogenous $A_T$ distribution along the Iberian coast (see Figure 2a in Cossarini et al., 2015). In fact, the linear $A_T$-SSS relationships presented by Cossarini et al., 2015 (see Figure 4 and Table 3) showed a lower correlation and greater residual dispersion in the eastern basin, indicative of greater variability compared to the western basin.

Following Cossarini et al., 2015, and due to the weaker influence of non-salinity factors on $A_T$ variation along the western boundary, we reconstructed $A_T$ using a salinity-based empirical relationship developed specifically for our study transect. The new equation presented in this paper (Eq. 1) aligns with the linear relationships proposed in various zones of this basin (Schneider et al., 2007, Copin-Montégut and Bégovic, 2002, Jiang et al., 2014, Cossarini et al., 2015).

Considering the limitations of the methodology applied for $A_T$ calculation, we have exercised caution in interpreting and inferring conclusions from results involving this variable. Although $A_T$ and $NA_T$ values are presented in Figures and Tables throughout the manuscript and supplementary materials, we have avoided direct discussion of these results. Instead, we used these results to support discussions related to the spatio-temporal variability of other physical and biogeochemical variables. In studying factors controlling $f\text{CO}_{2,\text{sw}}$ seasonality, Takahashi et al., 2014 applied Taylor decomposition (Eq. 6) to examine drivers of seasonal changes in $p\text{CO}_2$, pH, and $\Omega$ using $A_T$ data reconstructed from salinity relationships. However, we normalized $A_T$ values prior to including them in the Taylor decomposition using the most recent equation (Eq. 7) provided by Pérez et al., 2021, and extending trying to remove collinearity between $A_T$ and SSS. In Figure 5, we represented seasonal changes in $f\text{CO}_{2,\text{sw}}$ due to changes in SSS $\left(\left(\frac{\partial pCO_2}{\partial SSS} + \frac{NC_T}{SSS_0}\frac{\partial pCO_2}{\partial C_T} + \frac{NA_T}{SSS_0}\frac{\partial pCO_2}{\partial A_T}\right)\frac{dSSS}{dt}\right)$ and in $A_T$ $\left(\frac{SSS}{SSS_0}\frac{\partial pCO_2}{\partial A_T}\frac{dNA_T}{dt}\right)$, demonstrating an arbitrary relationship between them across all locations of interest during the study period. Moreover, seasonal changes in $p\text{CO}_2$ explained by SSS and $A_T$ are often below 5 μatm, significantly lower than those induced by SST and DIC (in agreement with Takahashi et al., 2014). This mean that changes in $f\text{CO}_{2,\text{sw}}$ explained by SSS and $A_T$ have minimal impact on the seasonal cycle of $p\text{CO}_2$ and are not crucial in quantifying changes driven by the two main drivers (temperature primarily, followed by DIC), which is the main conclusion we infer from this analysis.

I would also like to know how SSS and TA, which are less accurate in Equation (10), affect the results of the Taylor expansion. The author should calculate the error and clarify whether the Taylor expansion results are significant or not.

We have reviewed the Taylor deconvolution applied. The most important issue, which was not explicitly stated in the previous version of the manuscript and may be the main source of controversy, is that we did not use Equation 10 to calculate the seasonal amplitudes of the variables controlling the $f\text{CO}_{2,\text{sw}}$ changes. After a series of discussions, we decided to avoid using this equation for that purpose for the reason you pointed out: primarily, SSS and $A_T$ do not show a high correlation with Equation 10 because they lack a pronounced seasonal cycle compared to the other variables. This is because they are influenced by a set of processes that can be regional or local and occur over much longer timescales, without being directly driven by seasonal factors. In this transitional coastal area of the Western Mediterranean, these processes mainly include evaporation/precipitation, river runoff, and geochemical interactions with the coast and marine sediments in shallow areas that promote mineral dissolution. Based on these premises, we can expect that SSS and AT play a less relevant and more variable role, both interannually and spatially, in the seasonal change of in $f\text{CO}_{2,\text{sw}}$ compared to SST and $C_T$.

Therefore, given that the temporal changes in SST, SSS, $A_T$, and $C_T$ in Equations 6 and 7 do not necessarily coincide with those of in $f\text{CO}_{2,\text{sw}}$ on an annual scale, we assumed the seasonal variation as the difference in each of these variables between the times when in $f\text{CO}_{2,\text{sw}}$ reaches its maximum and minimum (this difference give the seasonal amplitudes), divided by the months elapsed. Seasonal amplitudes were calculated between monthly means (based on observations and computed data, not estimated through Equation 10) for

February and September (where minimum and maximum $f\text{CO}_{2,sw}$ were observed). An error propagation based on standard deviations was performed to calculate the uncertainty of associated with the difference between February and September means for each term and year. The results of solving Eq. 7 are presented in Figure 5 and the uncertainties associated to the seasonal changes of each term are shown in Table Sup 3.

The Taylor expansion applied to the seasonal cycle of $f\text{CO}_{2,sw}$ has been validated by direct comparison between the seasonal rate of change in $f\text{CO}_{2,sw}$ $\frac{df\text{CO}_{2,sw}}{dt}$ derived from the sum of each individual component $(\frac{\partial f\text{CO}_{2,sw}}{\partial X}\frac{\partial X}{dt})$ in Eq. 6 and 7, referred to in the manuscript with the subscript "sum," and the rate derived from the difference between the monthly averages of $f\text{CO}_{2,sw}$ observations in February and September, which we assume as the seasonal reference value for $f\text{CO}_{2,sw}$ and denote with the subscript "obs." Both seasonal rates are presented with their uncertainties, calculated via error propagation (Figure 5, Supplementary Table 3). The $\frac{df\text{CO}_{2,sw}}{dt}_{(sum)}$, considering its uncertainty range, falls within the uncertainty range of $\frac{df\text{CO}_{2,sw}}{dt}_{(obs)}$ (see errorbars in Figure 5). Additionally, a high degree of concordance is observed between both values, particularly across section S in the Alboran Sea. This consistency is highlighted in the manuscript and confirms the robustness of the seasonal change values obtained, thereby installing confidence in the methodology applied.

In the revised manuscript, we provide a much more detailed discussion of the methodology and its robustness to ensure clarity for readers and to facilitate an accurate interpretation of the results.

We have included in section 2.3.3 the following paragraphs:

"The seasonal changes of each driver (SST, SSS, $C_T$ and $A_T$) in Eq. 7 $\left(\frac{dX}{dt}\right)$ were assumed as their difference between the times of the year in which fCO2,sw was at its minimum and maximum (seasonal amplitudes) per months elapsed. Seasonal amplitudes were calculated between monthly means (based on observations and computed data) for February and September (where minimum and maximum $f\text{CO}_{2,sw}$ were observed). An error propagation based on standard deviations for February and September was performed to calculate the error of the seasonal change."

The first paragraph of section 4.4 was also modified as follows:

"To infer the causes of variations in the seasonal cycle of $f\text{CO}_{2,sw}$ among the study period, the seasonal rates of change of $f\text{CO}_{2,sw}$ $(\frac{df\text{CO}_{2,sw}}{dt}$, hereinafter $df\text{CO}_2)$ were decomposed into their individual components $(\frac{\partial f\text{CO}_{2,sw}}{\partial X}\frac{\partial X}{dt}$, hereinafter $df\text{CO}_2{}^X)$ as described in section 2.3.3 (Eq. 6 and 7). The results of solved Eq. 7 for each year at S1-S5 and E1-E6 are depicted in Figure 5. The positive values indicate an increase in $f\text{CO}_{2,sw}$ from February to September, while negative values the opposite. The uncertainty associated with the difference between the monthly means for each term and year was obtained through error propagation considering their individual standard deviations and presented in Table Sup 3. The $df\text{CO}_2$ resulted from the cumulative sum

of the individual terms in Eq. 7 (indicated with subscript "sum") matched the $dfCO_2$ directly calculated from observations between both seasons (indicated with the subscript "obs"), which renders confidence to the methodology (Figure 5)."

The discrete description of the study area is difficult to understand for those who are not familiar with this area. Thus, a "Study area" or similarly named subchapter should be added in Chapter 2 to describe the contents at Line 56-124, the hydrographical conditions, and previous studies of carbonate observations.

We agree that this study area has specific hydrodynamic characteristics that can be challenging for those unfamiliar with it, necessitating a detailed description. Following the journal's template and structure, we initially decided to include this description in the introduction, as observed in other regional/local scale articles published in Biogeoscience. We have now created a new subsection in the methodology titled "Study Area," which includes a description of the study area and the properties that may serve as potential sources of variability. This change has resulted in a much more concise and organized introduction and facilitate the comprehension of the entire study.

Line 287 Since river water does not reach zero alkalinity even with zero salinity (see Friis et al., 2003), the effect of river cannot be excluded with this method.

We have reviewed this point and agree we did not express it correctly. The traditional normalization method we are employing, which has also been utilized in prior studies applying Taylor deconvolution as outlined in Section 2.3.3, effectively removes the influence of salinity. This approach eliminates the effects of primary surface salinity-altering processes, such as evaporation, precipitation, and freshwater fluxes. However, this normalization still accounts for the influence of transport processes, including also vertical mixing and lateral advection, which can impact $A_T$ and $C_T$ concentrations.

In the specific case of river discharge, this normalization effectively removes the changes in $A_T$ and $C_T$ expected due to the decrease in salinity caused by freshwater input from river runoff near the mouth. However, we still account for changes in $A_T$ and $C_T$ that result from the input of dissolved carbonate minerals and bicarbonate-carbonate species.

Therefore, this normalization allows for the direct comparison of $A_T$ and $C_T$, as well as their influence on changes in $fCO_{2,sw}$ across the region. It also enables the identification of areas where advection and/or river discharges become significant and introduce modifications to the seasonal cycle of $fCO_{2,sw}$, which aligns with the objectives of this study.

**Minor comments**

Line 56 Abbreviations that appears for the first time in the maintext should be explained in the maintext.

Done in the new version of the manuscript.

Line 187 Is the error in the instrument itself or is it due to the temperature difference between the ocean and the intake?

It is the error in the instrument itself, in the new version of the manuscript is indicated as "instrumental error".

Line 226 One extra comma.

Typo. Removed in the new version of the manuscript

Line 295 What is different between S and SSS?

There was a typo in Eq. 7. As the study focuses on the surface, the salinity data is referred to throughout the manuscript as Sea Surface Salinity (SSS). Therefore, S was replaced with SSS in Eq. 7. We also noticed some inconsistencies in the explanation of the normalization procedure for $A_T$ and $C_T$ in the previous lines due to potential discrepancies in the terminology, and that $SSS_0$ was not defined in the text.

We have reformulated these lines as follows:

"…the most recent equation (Eq. 7) given by Pérez et al., (2021) with salinity-normalized $C_T$ and $A_T$ ($NC_T$ and $NA_T$) was used. The normalization was performed to a constant salinity ($SSS_0$) of 37.4 ($NX_T = SSS_0 * X_T / SSS$), which is the average SSS for the entire monitored area".

Line 318 The first paragraph of this chapter should be moved to the beginning of chapter 2.3. Also, if equation (10) is being applied to the data used in equations (6)-(9), it should be listed before those equations.

We have moved this paragraph to section 2.3.1. In the updated version of the manuscript, section 2.4 titled "Data adjustments and statistical procedures" only includes the procedure used for studying seasonal and interannual variability. In the case of Equation 10, it is applied to the average values along each route completed by the CanOA-VOS for each of the measured and computed variables. This procedure aim of study the temporal evolution of each variable and describe seasonal cycles that fits the observations. This equation was not used for calculating the drivers of $f$CO$_{2,sw}$ seasonality (Section 2.3.3, Eq. 6 and 7), but it was applied to the FCO$_2$ values calculated as described in Section 2.3.4 (Eq. 8 and 9). Therefore, after careful consideration, we believe that Equation 10 is listed in the text in a position consistent with the data processing workflow and the topic of section 2.4.

Line 399 Is this mean that the 11 points were determined by the seasonal variation of related parameters? If so this section should be moved to the second paragraph of the Result.

No, the selection of stations was based on the location of points of interest along sections S and E that were potential sources of variability. In making this selection, we considered factors such as the hydrodynamics, proximity to geographic features such as gulfs and capes where surface currents diverge and/or recirculate, and the bathymetry. The variability in seasonal amplitude across the two sections is already visible in Figure 2, as shown by the spatial differences between winter and summer, and is further analysed through the strategic positioning of these stations. We have explained that in a better way in the new version.

Line 557 The description about the SST reanalysis data should be moved to the Material and Method.

Done. The third paragraph in section 2.2 was modified as follow:

"The sea surface temperature (SST, in ºC) was monitored by using a SBE38 thermometer placed at the primary seawater intake in the engine room, with a reported instrumental error of ±0.01ºC. The high sensitivity of $xCO_2$ to temperature fluctuations required to measure the temperature at different locations along the system. A SBE45 thermosalinograph and a Hart Scientific HT1523 Handheld Thermometer, with reported instrumental errors of ±0.01ºC, were used to monitor the temperature at the entrance of the wet box and inside the equilibrator, respectively. The measured SST was analysed in conjunction with SST reanalysis monthly data (0.042º x 0.042º; with dates spanning 24 years within 01/01/2000 and 01/03/2024) from the Med MFC physical multiyear product (Escudier et al., 2020; 2021; Nigam et al., 2021), available at Copernicus Marine Data Store (https://data.marine.copernicus.eu/products). The SST reanalysis data was interpolated to the coordinates of the CanOA-VOS data to perform direct comparison in their dynamics."

Line 657 The description should be moved to the Material and Method, or the Result.

We have moved some descriptive lines into section 2.3.2.

Figure 2 It would be better to make the fCO2 and pH graph in Fig. 2 the Supplementary and nCT and nAT in the maintext.

Initially, we considered including the $NA_T$ and $NC_T$ plots alongside the other variables in Figure 2. However, this would have resulted in a figure with excessive information that could distract the reader. Since the main focus of the text is on the changes in $fCO_{2,sw}$ (and pH) in relation to SST changes, we deemed the inclusion of these variables in Figure 2 to be essential. Given the non-thermal processes that occur in the area and introduce spatial differences, we included a fourth plot for $C_T$, which supports the discussion of the results for the other variables, though it is not a primary focus of this article. Due to its lesser relevance in the discussion, but considering its importance in presenting these results to inform the reader about the variations in the other MCS variables, we decided to include the $A_T$, $NA_T$ and $NC_T$ plots in the supplementary material.

**Point-by-point response to Reviewer 2**

**General comment**

The topic is interesting, and the authors have worked a lot on several aspects of the marine carbonate systems and air – sea $CO_2$ fluxes in the study area. The methodologies and analysis follow robust techniques which help to support the results and discussion. The analysis of the trends shows how significant the observed trends are however a 6-year period is rather short and an inherent limitation. This is acknowledged by the authors and the high statistical significance is a good indicator of the validity of the assumptions. The authors do compare some of their findings (e.g. SST trends) to others, however, the comparison has limitations as they are made against datasets/products that cover the entire Mediterranean Sea and in some cases areas that are in the Atlantic (the ESTOC site is not very far but still the physical characteristics should be distinctively different). Since the main finding of this work is the strong relationship between SST increase and subsequent changes in CO2 fluxes and MCS, a more concise SST trends analysis will help the document.

We sincerely thank you for your thoughtful and constructive feedback on our manuscript. We greatly appreciate your recognition of the relevance of our study. Your comments have been carefully considered and have played a key role in improving the overall quality, clarity, and robustness of the revised manuscript. Below, we provide a point-by-point response to each of your comments.

The authors haven't investigated any other data sources that can be used as quality control for their data and/or gap filling. As an example, a simple search in SOCAT shows that there are 90 datasets in the area from 2010 until 2023 and 27 datasets from 2019 – 2023.

As an initial step in our data analysis, we performed a quick comparison with SOCATv2024 data (Bakker et al., 2016, 2024), as we have done in previous studies that used data from the ES-SOOP-CanOA station (Curbelo-Hernández et al., 2021a, 2021b). However, given the low data availability in this region in SOCATv2024 relative to the high-frequency ES-SOOP-CanOA dataset, we chose not to include explicit statements about this comparison in the final version of our manuscript. Your comment has made us realize the importance of justifying this point for readers and future users of the ES-SOOP-CanOA data.

In the revised version of the manuscript, we have added new statements regarding this comparison in Section 3 and included a new figure in the Supplementary Material (now Figure Sup 2). We found that the total number of $fCO_{2,sw}$ data collected over the five-year observation period (2019–2024) by ES-SOOP-CanOA (157,984 data points) exceeds the total number of historical data available in the Western Mediterranean (34.8–43.1ºN, 5.5ºW–4.7ºE) since 1999 in SOCATv2024 (146,094 data points; 44,520 when considering only the 2019–2023 period). The largest data gap in the ES-SOOP-CanOA dataset (one year) occurred between September 2021 and September 2022 due to the temporary cessation of the measurement system for vessel maintenance in dry dock. In an initial step of data processing, we attempted to fill this gap using SOCAT v2024 data. However, we found that SOCATv2024 also displays a one-year gap during this same period (Figure Sup 2).

Apart from this, the ES-SOOP-CanOA dataset provides a much higher spatiotemporal resolution than SOCATv2024 (Figure Sup 2), with data gaps generally shorter than one month, mostly due to technical issues with the measurement equipment that were

addressed during routine maintenance visits to the vessel. The total number of data points in each dataset for sections S and E is also shown in Figure Sup 2.

In the Alboran Sea (Section S; Figure Sup2b), $fCO_{2,sw}$ values from ES-SOOP-CanOA are consistent with those in SOCATv2024, although the limited number of cruises covering this section in SOCATv2024 difficult a direct comparison and prevent robust characterization of spatial and seasonal variability patterns. Along the eastern Iberian margin (Section E; Figure Sup2a), some differences between the two datasets are observed (i. e. during spring–summer 2021, $fCO_{2,sw}$ was higher in SOCATv2024 than in the ES-SOOP-CanOA dataset). These differences are mainly explained by the distinct sampling trajectories in SOCAT v2024, with some routes extending further eastward, including coastal areas around the Balearic Islands.

[Figure]

Figure Sup2. Spatio-temporal distribution of $fCO_{2,sw}$ data collected by the ES-SOOP-CanOA station and available at the SOCATv2024 dataset within 2019-2024 in (a) Section E and (b) Section S. $n$ refers to the total number of data points or each section and dataset.

The use of English is average, and in some cases it's difficult to follow sentences. There are numerous cases where grammar, syntax, words, entire sentences and small sections need to be edited/changed.

We have carefully revised the entire manuscript to improve clarity, grammar, and overall readability. We have restructured several sentences and sections to ensure clearer and more concise scientific communication.

There are good elements in this work which are worth publishing. Reducing the size and trying to focus on a specific question (e.g. CO2 fluxes, with only the minimum input of MCS or vice versa) improving the use of English and reducing the size will also help. A lot of work is done but still some work is needed to refine the document.

We thank the reviewer for this positive and constructive feedback. As suggested, we have revised the manuscript to improve its focus, reduce its length, and enhance the overall

clarity and use of English. We have thoroughly discussed the main objectives and perspectives of the article. Given that the article introduces a novel database for the Western Mediterranean, we consider essential to delve into methodological aspects related to data acquisition and processing (Section 2), as well as the spatiotemporal distribution of the results (Section 3). These sections lead into a discussion (Section 4) where we perform additional analyses to investigate the dynamics of the MCS across temporal scales ranging from seasonal to interannual, including an analysis of air-sea $CO_2$ fluxes. These derived aspects are presented in independent subsections within the *Discussion* (Section 4). We have also modified the text and structure following the suggestions by Reviewer 1. The main structural changes implemented in the revised manuscript are as follows:

- Part of the descriptive content about the study region originally included in the Introduction and Discussion has been moved to a new subsection in the Methodology titled *Study Area* (Section 2.1).

- The former section *Data Adjustments and Statistical Procedures* (Section 2.3.1 in the previous version) has been moved to *Appendix A* to streamline the main text. Additionally, we have included *Appendix B*, which provides a detailed description of the uncertainty estimation for the air–sea $CO_2$ fluxes associated with uncertainties in $f\mathrm{CO}_{2,sw}$ and $f\mathrm{CO}_{2,atm}$.

- Several structural aspects of the *Methods* and *Results* sections have been improved, including the merging of the previous two results subsections into a single Section 3.

- The structure and narrative of the *Discussion* have been substantially revised. Subsections 4.1.1 and 4.1.2 have been introduced under Section 4.1 to better organize the discussion of the results according to the two distinct study regions (Alboran Sea and eastern Iberian margin).

- The *Conclusions* section has been shortened, restructured, and clarified to provide a more concise and effective closing to the manuscript.

We believe these changes have resulted in a more coherent, focused, and accessible manuscript, while maintaining the scientific integrity and depth of the study.

**Specific comments on sections**

**Abstract**

Line 16: The 45-83% referring to the fCO2 or the SST? As "fluctuation", do you mean variability? If there's an offset, then what's the final seasonal magnitude? A bit confusing.

We have clarified this point in the revised abstract, which we include below in response to the following comment.

The statement in the abstract that the CO2 sink has reduced 40-80% since 2019, is strong, however 40 – 80 % is a very large margin and it will be more concise if there is a (very) brief explanation for this margin.

The wide range between 40% and 80% is due to the large spatial variability in $CO_2$ fluxes across the study region. We have added a brief clarification in the abstract, which we include in full below:

"The surface physical and Marine Carbonate System (MCS) properties were assessed along the western boundary of the Mediterranean Sea. An unprecedent high-resolution observation-based dataset spanning 5 years (2019-2024) was built through automatically underway monitoring by a Volunteer Observing Ship (VOS). The MCS dynamics were strongly modulated by physical-biological coupling dependent on the upper-layer circulation and mesoscale features. The variations in $CO_2$ fugacity ($fCO_{2,sw}$) were mainly driven by sea surface temperature (SST) changes. On a seasonal scale, SST explained 45-83% of the increase in $fCO_{2,sw}$ from February to September, while total alkalinity ($A_T$) and sea surface salinity (SSS) explained <15%. The processes controlling total inorganic carbon ($C_T$) partially offset this increment and explained ~25-38% of the $fCO_{2,sw}$ seasonal change. On an interannual scale, the SST trends (0.26-0.43 ºC $yr^{-1}$) have accelerated by 78-88% in comparison with previous decades. The ongoing surface warming contributed by ~76-92% in increasing $fCO_{2,sw}$ (4.18 to 5.53 µatm $yr^{-1}$) and, consequently, decreasing pH (-0.005 to -0.007 units $yr^{-1}$) in the surface waters. The seasonal amplitude of SST, becoming larger due to progressively warmer summers, was the primary driver of the observed slope up of interannual trends. The evaluation of the air-sea $CO_2$ exchange shows the area across the Alboran Sea (14,000 $Km^2$) and the eastern Iberian margin (40,000 $Km^2$) acting as an atmospheric $CO_2$ sink of -1.57 ± 0.49 mol $m^{-2}$ $yr^{-1}$ (0.97 ± 0.30 Tg $CO_2$ $yr^{-1}$) and -0.70 ± 0.54 mol $m^{-2}$ $yr^{-1}$ (-1.22 ± 0.95 Tg $CO_2$ $yr^{-1}$), respectively. Considering the spatial variability of $CO_2$ fluxes across the study area, a reduction of approximately 40–80% in the net annual $CO_2$ sink is estimated since 2019, which is attributed to the persistent strengthening of the source status during summer and the weakening of the sink status during spring and autumn."

**Introduction**

In line 63 it is mentioned that the NW Med is a relatively weak sink for atm $CO_2$, while in line 70 it is mentioned that the same area has a strong atm $CO_2$ uptake. A bit confusing.

Thank you for noticing, the previous sentence was somewhat unclear. We have revised it as follows:

"Long-term variations in MCS within the northwestern Mediterranean occur at rates exceeding those anticipated from chemical equilibrium with atmospheric $CO_2$, which has been attributed to the intense deep-convection processes in this area (Copin-Montégut, 1993; D'Ortenzio et al., 2008; Cossarini et al., 2021) and the substantial input of anthropogenic carbon from the North Atlantic (Merlivat et al., 2018; Palmiéri et al., 2015; Schneider et al., 2010; Ulses et al., 2023)".

Line 105: what does exitance mean? Is it exit, is it existence?

Thank you for noticing the typo. It is *existence* and was corrected in the revised version.

Line 126: The sentence "an alternatively …" doesn't make sense. Why is the methodology alternative? It's robust and powerful but alternative to what?

We reformulated the entire last paragraph of the *Introduction*:

"This research focus on the surface spatio-temporal variations of the MCS and air-sea $CO_2$ fluxes in the western boundary of the Mediterranean Sea. High-resolution and reliable data were obtained through autonomous underway monitoring of the surface ocean from February 2019 to February 2024 by a Volunteer Observing Ship (VOS). This systematic strategy represents a powerful tool to analyse the distribution and

changes of physical and MCS properties in highly variable areas as coastal transitional zones where the availability of data has been historically scarce. The cruise track (Figure 1) followed the south and east geographically rugged coastline of the Iberian Peninsula and allowed the characterization of the Alboran Sea (~2-5.1ºW) separately from the eastern coastal and shelf area between Cape of Gata (Almería) and Barcelona (~36.5-41.3ºN). The changes observed in the MCS on a seasonal and interannual timescales (even considering the limitations of 5 years of data), the mechanism controlling their variations and the changes in the air-sea $CO_2$ exchange have been attended in this study."

Line 140: Improve our knowledge of what?

We agree that the last line of the *Introduction* was a bit confusing. We have reformulated it (see comment above).

**Methods**

Minor comment: Can the authors clarify whether the VOS is operating under the WMO / GOOS network? The term SOOP in the document is for Surface Ocean Observation Platform, yet in ICOS this is defined as a Ship Of Opportunity? Acronyms are very confusing and might be necessary to use established ones.

Thank you for noticing the mistake. Some statements in section 2.1 (2.2 in the new version of the manuscript) were modified and reorganized, and the ICOS established acronyms (https://www.icos-cp.eu/about/icos-in-nutshell/abbreviations; last access: 15 May 2025) are now used. The revised first paragraph in Section 2.1 (now Section 2.2) is included as follows:

"A high spatio-temporal resolution dataset spanning 5 years was constructed based on weekly physico-chemical observations of the surface western boundary of the Mediterranean Sea between February 2019 and February 2024. Data was automatically collected by the Volunteer Observing Ship (VOS) MV JONA SOPHIE (IMO: 9144718, called RENATE P before November 2021), a container ship managed in Spain by Nisa Maritima which links the Canary Islands with Barcelona. This VOS line was designed and is maintained by the QUIMA research group at the IOCAG-ULPGC, and operates within the framework of the Integrated Carbon Observation System (ICOS; https://www.icos-cp.eu/; last assess: 15 May 2025) as a Ship-of-Opportunity (SOOP) Ocean station (Station ID: ES-SOOP-CanOA) since 2021 (upgraded to an ICOS Class 1 Ocean Station in May, 2024). Therefore, the measurement equipment and underway data collection techniques verify the ICOS high-quality requirements and methodological recommendations."

Line 167: What is defined as "low atmosphere"? How high was the intake.

The air intake was placed at approximately 8 meters above sea level on the ship's second deck, outside the influence of the smoke plume and other potential contamination sources from the vessel itself. The seawater intake was at approximately 5 m depth. These data was added to the first lines of section 2.2 (2.3 in the new version of the manuscript).

What is the actual performance of the NDIR on the 2 modes (seawater and atmospheric)?

According to the manufacturer's specifications, the nominal accuracy of the LI-COR infrared gas analyzer is 1% for $CO_2$ concentrations within the range of 0 to 3000 ppm. By automatically calibrating the instrument (every 3 hours and each time the vessel docks at

a port) with standard gases with an accuracy of ±0.02 ppm and ranging from 0 to 800 ppm (which encompasses the measurement range of 300 to 600 ppm), performing in-loop zero and span calibrations every twelve hours and actively minimizing temperature and pressure drift through its continuous monitoring (with accuracies of ±0.01ºC and ±0.0002 atm), the system achieved the target accuracy of ±0.2 µatm for $f\mathrm{CO_{2,atm}}$ and ±2 µatm for $f\mathrm{CO_{2,sw}}$ (Pierrot et al., 2009). We have included a detailed description of these accuracies in the new Sections 2.3 and 2.4.1.

Have you compared the $f\mathrm{CO_{2,ATM}}$ with any gridded products or from different sources/stations (e.g. ATM station nearby, satellite products, Jena Carboscope, NOAA OCADS,…). Differences are to be expected but since other studies are using this data it might be useful to explain and justify the data used in this study. The work on fluxes requires this level of scrutiny.

As part of the initial stage of our data processing workflow, we routinely compare our $x\mathrm{CO_2}$ measurements in atmosphere ($x\mathrm{CO_{2,atm}}$) with $x\mathrm{CO_{2,atm}}$ data from the Izaña Atmospheric Research Center (IZO site; Tenerife, Canary Islands, Spain). This site serves as the reference station for the Northeastern Atlantic and is operated by the Spanish Meteorological Agency (AEMET). It is part of several major international atmospheric monitoring networks, including the Global Atmosphere Watch (GAW) programme of the World Meteorological Organization (WMO), the Network for the Detection of Atmospheric Composition Change (NDACC), and the NOAA GML Carbon Cycle Cooperative Global Air Sampling Network. For this study, we used daily dry air $x\mathrm{CO_2}$ data collected at the IZO site since 1991, available through the NOAA Global Monitoring Laboratory (GML) dataset [https://gml.noaa.gov/data/dataset.php?item=izo-co2-flask; last access: 14 May 2025]. In the revised manuscript, we have included a justification for the use of our $x\mathrm{CO_2atm}$ data in Section 2.3, and the following new figure in the Supplementary Material (now Figure Sup1) that illustrates this comparison.

[Figure]

Figure Sup 1. (a) $x\mathrm{CO_{2,atm}}$ measurements from ES-SOOP-CanOA (red) and Izaña Atmospheric Research Center (IZO site; blue) during 2019-2024. The IZO site (https://gml.noaa.gov/dv/site/site.php?code=IZO, last access: 14 May 2025) is located in Tenerife (Canary Islands, Spain; 28.3090ºN, 16.499ºW) and placed at 2372.9 m above sea level. Dry air $x\mathrm{CO_2}$ data collected at the IZO site are available through the NOAA GML dataset (https://gml.noaa.gov/data/dataset.php?item=izo-co2-flask; last access: 14 May 2025). (b)

Differences between $x\mathrm{CO}_{2,\mathrm{atm}}$ from ES-SOOP-CanOA and $x\mathrm{CO}_{2,\mathrm{atm}}$ from IZO ($\Delta x\mathrm{CO}_{2,\mathrm{atm}}$), with a histogram illustrating the statistical distribution and normality of the differences.

We found that our $x\mathrm{CO}_{2,\mathrm{atm}}$ measurements are consistent with those from the IZO station, despite spatial differences in the respective sampling locations. During the period 2019–2024, our $x\mathrm{CO}_{2,\mathrm{atm}}$ data were, on average, 1.14 ppm higher than those recorded at IZO. This offset can be attributed to the fact that air sampling at IZO is conducted at approximately 2400 meters above sea level, in a remote location far from major urban or industrial areas and above the atmospheric inversion layer, which shields the station from surface-level pollution. In contrast, the ES-SOOP-CanOA measurements are conducted in the lower atmosphere, near the sea surface and closer to greenhouse gas emission sources (particularly when the vessel operates near the coast along inter-island routes in the Canary Islands or between the Strait of Gibraltar and Barcelona in the Mediterranean basin).

Line 196: It will be informative if the "difference in the order of millibars" is better defined as this can be a crucial source of error for the final $f\mathrm{CO}_2$.

We reformulated these statements. The system measures the pressure at the deck box transducer close to the air intake (atmospheric pressure), at the wed box inside the equilibrator at the time of equilibration and in the dry box inside the LI-COR. Minimal pressure changes do not represent a significant source of error in $f\mathrm{CO}_{2,\mathrm{sw}}$ measurements, since the $x\mathrm{CO}_2$ values obtained in the engine room were corrected to in situ conditions by following the calculation procedure given by Pierrot et al., (2009). We calculated first the partial pressure of $\mathrm{CO}_2$ at equilibrium conditions (at the temperature and pressure measured inside the equilibrator and considering that the $\mathrm{CO}_2$ concentration in the headspace gas inside the equilibrator is proportional to the $\mathrm{CO}_2$ concentration in seawater when equilibrium is reached [Henry's law]; see Equation 4 in Pierrot et al., 2009). Our next step was to calculate the $\mathrm{CO}_2$ fugacity at equilibrator temperature and at atmospheric pressure (see Equation 1 in Pierrot et al., 2009). Lastly, we corrected the $f\mathrm{CO}_{2,\mathrm{sw}}$ to "in situ" conditions by using the empirical temperature dependence given by Takahashi et al., 1993 (see Equation 6 in Pierrot et al., 2009).

It's not clear where the $A_T$ and $C_T$ data from the discrete seawater samples are used. I suspect that the $A_T$ discrete water samples were used to produce equation 1, but where are the $C_T$ discrete water samples used? Are they used in Fig 2(?) and or Fig Sup 4? Are they used to "check" the calculated $C_T$?

Discrete seawater samples were analysed using a VINDTA 3C for the determination of $A_T$ and, as an additional variable, $C_T$. $A_T$ was used to establish the relationship with SSS (Eq. 1). $C_T$ values were only used in an initial data processing step to assess the consistency between computed and experimentally determined values. Both values were consistent within a range of $\pm2$ µmol kg$^{-1}$.

Line 227: $C_T$ is not determined via titration but usually by adding acid and stripping the CO2 from the sample and measuring the amount stripped in a coulometer or CO2 detector.

We agree with the reviewer. $C_T$ was analysed by coulometry using the VINDTA 3C, as in our previous studies (Curbelo Hernández et al., 2021a, 2021b). Since these data were only used in an initial step of the data processing to validate the consistency of the computed values (see comment above), and were not used for further calculations, discussion of

results, or drawing conclusions, the description of the $C_T$ determination procedure has been removed from the revised version.

Line 237: The reported error of 17.1 umol/kg is very large and limits the use of $A_T$ as a parameter for calculating other parameters in the carbonate system (check this documented by T. Steinhoff; Uncertainty analysis for calculations of the marine carbonate system for ICOS-Oceans stations; doi: 10.18160/vb7c-z758).

We refined Eq. 1 by removing outliers and recalculated $A_T$ at the time of each $f\mathrm{CO}_{2,sw}$. A new equation was then constructed based on 46 measurements, yielding an R² of 0.99 and an RMSE of 5.7 µmol kg$^{-1}$. The slope and intercept coefficients of this new equation fall within the error intervals of those from the previous Eq. 1. Differences between the previous and new $A_T$ estimates were negligible (<0.1 µmol kg$^{-1}$). The propagated uncertainty in $A_T$ estimates, accounting for instrumental errors in AT and SSS measurements (1.5 µmol kg$^{-1}$ and 0.0005, respectively) and the linear model uncertainty, was approximately ±5.7 µmol kg$^{-1}$. This error in $A_T$ estimation now falls within the accepted uncertainty range of ±10 µmol kg$^{-1}$ for $A_T$ when used as an input variable alongside $f\mathrm{CO}_{2,sw}$ (when its uncertainty is up to ±2 µatm) for the calculation of other MCS variables (see Figure 3 in Steinhoff and Skjelvan, 2020). Considering the observed range of SST (14–28°C) and $f\mathrm{CO}_{2,sw}$ (300–570 µatm), the uncertainty in the calculation of $C_T$ and pH is estimated to be less than ±9.45 µmol kg$^{-1}$ and ±0.0059 units, respectively (Steinhoff and Skjelvan, 2020). These uncertainties remain below the thresholds of ±10 µmol/kg for $A_T$ and $C_T$, ±2 µatm for $f\mathrm{CO}_{2,sw}$ and 0.02 units for pH, which align with the criteria for the "weather goal" level of measurement quality (Steinhoff and Skjelvan, 2020). This level is defined as sufficient to resolve relative spatial patterns and short-term variability, as well as to support mechanistic understanding and responses to local and immediate ocean acidification dynamics.

In the revised version of the manuscript, we have modified the paragraph related to the explanation of the determination of $A_T$ in Section 2.3.1 (now Section 2.4.1), according to these updates along with those made after considering the comments from Reviewer 1.

The revised paragraph is included below:

> The discrete seawater samples were analysed for $A_T$ by using a VINDTA 3C and following the procedure detailed by Mintrop et al., (2000). The VINDTA 3C was calibrated through the titration of Certified Reference Material (CRMs; provided by A. Dickson at Scripps Institution of Oceanography), giving values with an accuracy of ±1.5 µmol kg$^{-1}$. A new approach to reconstruct $A_T$ using salinity-based empirical relationship was built specifically for the monitored transect. The $A_T$-SSS linear relationship obtained from 46 discrete samples (Eq. 1) is statistically significant at the 99% level of confidence (p-value < 0.01) and present a high degree of correlation ($r^2$= 0.99) and a RMSE of ±5.6 µmol kg$^{-1}$. The propagated uncertainty in $A_T$ estimates, considering the errors in $A_T$ determination and SSS measurements (Section 2.3) and the linear model uncertainty, was approximately ±5.7 µmol kg$^{-1}$. This error in $A_T$ estimation now falls within the accepted uncertainty range of ±10 µmol kg$^{-1}$ for $A_T$ when used as an input variable alongside $f\mathrm{CO}_{2,sw}$ (when its uncertainty is up to ±2 µatm) for the calculation of other MCS variables aligning with the criteria for the "weather goal" level of measurement quality (Steinhoff and Skjelvan, 2020). This linear relationship aligns with those proposed in various zones of the Mediterranean Sea (Schneider et al., 2007, Copin-Montégut and Bégovic, 2002, Jiang et al., 2014, Cossarini et al., 2015). Although the reconstruction of $A_T$ from its linear relationship

with SSS does not account for biological processes that cannot be traced with salinity (Wolf-Gladrow et al., 2007), nor the input of dissolved carbonate minerals and bicarbonate-carbonate species from river runoff, sediments, and water mixing, it has been widely used and provides a useful general approximation in regions with stable conditions and less influenced by these processes. Considering that the influence of biological cycles on $A_T$ is reduced along the western boundary of the Mediterranean Sea due to the influx of cooler and nutrient-rich Atlantic waters, and that terrestrial and riverine contributions have minimal influence on $A_T$ distribution compared to marginal and coastal areas in the Eastern Mediterranean Basin (Cossarini et al. 2015), the $A_T$ was calculated at the times of the observations (Curbelo-Hernández et al., 2021a; 2021b; 2023) using Eq. 1. This new $A_T$-SSS relationship can be used to calculate the $A_T$ content in surface seawaters subject to low influence of non-salinity factors in the western Mediterranean Sea, with salinities ranging between 36 and 38.5.

$$A_T = 100.5 \, (\pm 2.9) \, SSS - 1271(\pm 108) \tag{1}$$

Reference:

Steinhoff, T., Skjelvan, I., 2020. Uncertainty analysis for calculations of the marine carbonate system for ICOS-Oceans stations. ICOS OTC. https://doi.org/10.18160/VB7C-Z758.

Line 277: The term $\Delta fCO_2$ is introduced but without explanation. I suspect it's the $fCO_{2,SW}$ - $fCO_{2,ATM}$ (?)

We already included the explanation in section 2.3.4 (now Section 2.4.4 in the revised version of the manuscript).

Section 2.3.4: Following from comments on the performance of the NDIR for both modes, and since errors/uncertainties in the $\Delta fCO_2$ term propagate and become significant, it will be beneficial if the authors elaborate on how the flux errors have been calculated.

This is an important point. To clearly convey it, we have restructured Section 2.3.4 (now Section 2.4.4 in the revised version of the manuscript) to include the uncertainties associated with each of the terms in the bulk formula in Eq. 8. We also included an Appendix explaining the calculation of the uncertainty in $FCO_2$ associated solely with the propagated error from $\Delta fCO_2$.

The new Section 2.4.4 is presented below:

2.2.4. Air-sea $CO_2$ fluxes

The air-sea $CO_2$ fluxes ($FCO_2$) were determined using the bulk formula (Broecker and Peng, 1983) in Eq. 8:

$$FCO_2 = 0.24 \, K_0 \, k_{660} \, \Delta fCO_2 \tag{8}$$

where $K_0$ is the solubility of $CO_2$ in seawater, $k_{660}$ is the gas transfer velocity and $\Delta fCO_2$ represents the difference between $fCO_{2,sw}$ and $fCO_{2,atm}$. A conversion factor of 0.24 was used to express $FCO_2$ values in units of mmol m$^{-2}$ d$^{-1}$. $K_0$ was calculated by

using the equation and coefficients given by Weiss, (1974) and measured SST and SSS which fall within the valid application limits. Considering the fitting error from the original parameterization of $K_0$ ($\pm 1 \times 10^{-4}$ mol L$^{-1}$ atm$^{-1}$; Weiss, 1974) and the instrumental errors of SST and SSS measurements (section 2.3), the uncertainty associated with the solubility estimation had a negligible impact on the calculation of FCO$_2$. $k_{660}$ was calculated through its quadratic dependency with wind speed (Eq. 9) using the parametrization given by Wanninkhof (2014):

$$K_{660} = 0.251 \cdot w^2 \cdot \left(\frac{Sc}{660}\right)^{-0.5} \qquad (9)$$

where $w$ is the wind speed and $Sc$ is Schmidt number (cinematic viscosity of seawater, divided by the gas diffusion coefficient). ERA5 hourly wind speed reanalysis data at 10 m above the sea level and with a spatial resolution of 0.25º x 0.25º (Hersbach et al., 2023) were used to calculate $k_{660}$. The ERA5 reanalysis for the global climate and weather is available at Copernicus Climate Data Store (https://cds.climate.copernicus.eu/). The uncertainty in $k_{660}$ reported by Wanninkhof (2014) when using wind speeds ranging between 3 and 15 m s$^{-1}$ is $\pm 20\%$. The error in the determination of $f$CO$_{2,\text{sw}}$ and $f$CO$_{2,\text{atm}}$ (Section 2.4.1) propagates into the calculation of $\Delta f$CO$_2$ and constitutes an additional source of uncertainty. The statistical procedure used to quantify the uncertainty in FCO$_2$ arising from the uncertainty in $\Delta f$CO$_2$ is described in Appendix B. The mean absolute error in FCO$_2$ due to the propagated uncertainty of $\Delta f$CO$_2$ ($\pm 2.01$ µatm) was $\pm 0.14$ mmol m$^{-2}$ d$^{-1}$, which in relative term is $\pm 0.05\%$. Negative FCO$_2$ values indicate that the ocean acts as an atmospheric CO$_2$ sink, while the positive ones indicate that it behaves as a source.

The Appendix B is provided below:

Appendix B: Uncertainty in FCO$_2$ explained by the propagated error in $\Delta f$CO$_2$

The uncertainty in $\Delta f$CO$_2$ was calculated by applying standard error propagation rules for the difference of two independent measurements with associated uncertainties (Eq. B.1):

$$\sigma_{\Delta f CO_2} = \sqrt{\sigma_{f CO_{2,sw}}^2 + \sigma_{f CO_{2,atm}}^2} \qquad \text{B.1}$$

where $\sigma_{f CO_{2,sw}}$ and $\sigma_{f CO_{2,sw}}$ are the uncertainties for $f$CO$_{2,\text{sw}}$ and $f$CO$_{2,\text{atm}}$, respectively (see section 2.4.1). The absolute error in FCO$_2$ ($\sigma_{FCO_2}$; mmol m$^{-2}$ d$^{-1}$) associated solely with uncertainty in $\Delta f$CO$_2$ was estimated for each data point using Eq. B.2:

$$\sigma_{FCO_2} = K_{660} \, K_0 \, \sigma_{\Delta f CO_2} \qquad \text{B.2}$$

To represent the average magnitude of uncertainty in the estimated FCO$_2$ over the entire dataset (with $n$ being the total number of data), the mean absolute FCO$_2$ error was calculated using Eq. B.3 and the mean relative FCO$_2$ was estimated with Eq. B.4:

$$\overline{\sigma_{FCO_2}} = \frac{1}{n}\sum_{i=1}^{n}\sigma_{FCO_2,i} \qquad\qquad\qquad B.3$$

$$\frac{\overline{\sigma_{FCO_2}}}{FCO_2} = \frac{1}{n}\sum_{i=1}^{n}\left|\frac{\sigma_{FCO_2,i}}{FCO_2}\right| * 100 \qquad\qquad\qquad B.4$$

**Results**

Line 358: Sentence starting from "Based on differences…" is confusing and difficult to follow. Please rephrase. Also what are the hydrodynamical processes and oceanographic features?

This sentence was reformulated in the new version of the manuscript.

Fig 3: Will be useful if the authors elaborate more on the poor fit of fCO2 in S1 and for pH in all regions.

It is true that this is an important point to comment. We have added this lines in the Discussion (Section 4.1.1):

"The relatively low SST and $fCO_{2,sw}$ around S1 (20.68 ± 2.20 ºC and 401.68 ± 27.13 µatm) and S3 (21.15 ± 2.11 ºC and 407.30 ± 26.20 µatm) were mainly due to the highest intensity of the wind-induced upwelling along the northern coast of the western Alboran Sea during the warm season. It cooled the surface and enhanced the biological drawdown (e. g. Bolado-Penagos et al., 2020; Folkard et al., 1997; Gómez-Jakobsen et al., 2019; Peliz et al., 2009; Richez and Kergomard, 1990; Stanichny et al., 2005), while favouring the formation of the cold and nutrient-rich filament separating the WAG and EAG (Gómez-Jakobsen et al., 2019; Millot, 1999). Differences in the influence and strength of this filament may contributed to the observed heterogeneities in SST, $fCO_{2,sw}$, and pH at S1 during the warm seasons (Figure 3), which in turn account for reducing the model fitting performance".

Both Fig 3 and 4 are crowded (especially the fCO2 plots that have many elements). Maybe show fco2 total and have the components in the supplementary section.

We agree that the figure, particularly the $fCO_{2,sw}$ panels, appears somewhat crowded. We had already considered alternative ways to present these data, but after careful evaluation, we believe that the current format is the most effective. It allows the reader to clearly assess the consistency between the thermal and non-thermal fCO2 components estimated using the T'02 (Takahashi et al., 2002) and F'22 (Fassbender et al., 2022) methods (e.g., the close overlap between orange and purple symbols and lines, as well as between green and blue; see Figures 3 and 4). Additionally, this layout makes it possible to visualize how both components contribute to the seasonal and interannual variability of $fCO_{2,sw}$ in a coherent and integrated manner.

If we were to plot these components separately, the key insights regarding methodological consistency and the relative contributions of the thermal and non-thermal signals to $fCO_{2,sw}$ variability would not be as clearly conveyed.

Fig 7: It will be informative if you can include error bars (at least on the annual trend).

Initially, we attempted to include error bars for both the trends and the data points, but the figure became visually overloaded. As an alternative, we added Supplementary Table 4, which reports the standard deviations of the data points, and we included in each panel of Figure 7 the trends (slopes) followed by their corresponding standard errors.

Surprised that there are no results for fluxes. They are presented in the discussion and at the same time discussed, which has its merits but makes the document inconsistent.

The structure of the manuscript was designed intentionally so that the Results section focuses on the direct spatiotemporal distribution of the observed variables and determined MCS variables. This approach follows recommendations from previous reviewers in related works, who advised avoiding the inclusion in the Results section of variables that are derived from further calculations, such as air-sea $CO_2$ fluxes or the drivers of $f$CO$_{2,sw}$ seasonality. These derived aspects are instead presented in dedicated subsections within the Discussion, where they are interpreted in the broader context of biogeochemical processes.

This structure aims to clearly separate observations from further calculations, thereby reinforcing the robustness and transparency of the observational dataset before using it as a basis for more integrative analyses and conclusions concerning spatiotemporal changes in the MCS and air-sea $CO_2$ fluxes.

Table 1 and 2: is the term "Ratio ºC yr$^{-1}$" the trend? If so why not call it "Trend"?

Yes, it is the trend. It was corrected in the new version.

**Discussion**

Line 581: The 2000-2019 warming is 0.1 and 0.06 ºC/yr for S and E section however in Table 1 I can see 0.03 and 0.05 respectively. Is it possible to clarify?

We realized that the discussion of SST trends was a bit extensive and potentially confusing. Since this topic does not fall within the main scope of the article, we have aimed to streamline and shorten this section, making it much easier for the reader to follow. References to several trends that were already presented in Table 1 have been removed from the text. We include the revised first two paragraphs of Section 4.2 below:

"The monitoring of the surface Western Mediterranean Basin allowed the identification of interannual trends for physical and MCS properties (Table 1 and 2). The SST increased at a rate of 0.38 ± 0.05 ºC yr$^{-1}$ in the S section and 0.30 ± 0.04 ºC yr$^{-1}$ in the E section. The rate of increase in SST locally intensified at S2 (0.50 ± 0.09 ºC yr$^{-1}$) may be due to the transport and accumulation of surface waters toward the core of the WAG. Its variability, migration and progressively collapse can also account for the rapid warming of the area (Sánchez-Garrido et al., 2013; Viúdez et al., 1998; Vélez-Belchí et al., 2005).

The SST trends based on ES-SOOP-CanOA data were of the same order of magnitude as those derived from reanalysis data for the period 2019–2024, but were one order of magnitude higher than the reanalysis-based trends for 2000–2019, indicating a reinforcement of sea surface warming by approximately 80–90% (Table 1). The ES-SOOP-CanOA data-based interannual SST trends were found to be reinforced during summer by 55.2% in the S section and by 32.4% in the E section compared to winter. The Northern Current cooling the northernmost part of the E section accounted to decelerate the warming in comparison to the S section. The ES-SOOP-CanOA data-based trends reported a cumulative increase in SST from 2019 to 2024 of 1.91 ± 0.26 ºC in the Alboran Sea (S section) and 1.52 ± 0.22 ºC along the eastern Iberian margin (E section). These cumulative increments were 48.3% and 34.94% respectively higher than those estimated for the global surface ocean from 1850-1900 to 2001-2020 (0.99 ± 0.12 ºC; IPCC, 2023). It aligns with projections from climate models for both

terrestrial and marine environments in the mid latitudes, particularly within the Mediterranean region, in consequence of human-induced global warming, which was detailed by Hoegh-Guldberg et al., (2018) in the AR6 Synthesis Report (IPCC, 2023)."

Line 586: "in warming." Of what? Atmospheric warming? General increase in sea surface temperature?

It was corrected to "sea surface warming". See comment above.

Line 589 - 594: Skeptical on the validity of this comparison. Basically you are comparing a local 5 year trend against a global and 2 decades product.

We agree that this comparison may cause confusion and potentially introduce noise in the overall narrative of the manuscript. For this reason, we have removed it from the revised version. See comments above.

Line 758: What does "isolated eventual improved injections" mean? Where do these waters come from?

There was a typo in this statement that we already corrected: "The lowest compensation found in 2019 at E5 (28.8%) and E6 (18.4%) was likely related with eventual injections of remineralized waters along the Northern Current path, which offset the biological uptake of $C_T$ and elevated the $dfCO_2^{CT}$".

Line 761: Very surprised and confusing to start with "The Eastern Boundary of the Med Sea…". Is this a typo? Is this the eastern part of the study area? Please clarify/correct.

We have revised this paragraph to provide a more appropriate starting of Section 4.5. The updated opening paragraph is as follows:

"The continuous observation of MCS variables enabled the calculation of $FCO_2$ at an unprecedented high spatiotemporal resolution in the Western Mediterranean Sea. The $FCO_2$ was found to be governed by fluctuations in $\Delta fCO_2$ (Figure 6), mainly controlled by the broader variability of $fCO_{2,sw}$ (325-500 µatm) compared to $fCO_{2,atm}$ (390-425 µatm). The SST variations have a relevant role by primary controlling $fCO_{2,sw}$ (section 4.3) and modulating the solubility of $CO_2$ at the air-sea interface. The entire monitored area was undersaturated for $CO_2$ respect to the low atmosphere between late October and June ($\Delta fCO_2$= -35.30 ± 8.97 µatm), acting as an atmospheric $CO_2$ sink (-2.56 ± 0.55 mmol m$^{-2}$ d$^{-1}$) which peaks in winter (-4.53 ± 0.44 and -3.29 ± 0.31 mmol m$^{-2}$ d$^{-1}$ in S and E sections, respectively). During summer, the area was supersaturated for $CO_2$ ($\Delta fCO_2$= 36.43 ± 0.35 µatm) and acted as a source, which was about three times more intense along the E section (1.70 ± 0.43 mmol m$^{-2}$ d$^{-1}$) compared to the S section (0.57 ± 0.35 mmol m$^{-2}$ d$^{-1}$)."

Line 861: The use of different units for the flux is confusing. For example, in line 849 the unit is mol/(m$^2$yr) and in is $TgCO_2$/yr.

The use of different units to present $FCO_2$ values was intentional and follows conventions widely used in the literature, enabling comparison with previous studies on air-sea $CO_2$ exchange at both regional and global scales. We begin Section 4.5 by discussing the temporal (seasonal and interannual) variability of $FCO_2$ using Figure 6, where fluxes are reported in mmol m$^{-2}$ d$^{-1}$. Since this analysis focuses on variations occurring over days to months, we considered it appropriate to express $FCO_2$ in daily units, which better reflect

the temporal resolution of the data and provide values that are more intuitive for the reader.

When we move on to evaluate the ingassing and outgassing for each season and the full year using seasonally and annually averaged data (Figure 7), we switch to mol m$^{-2}$ yr$^{-1}$, as these units are more suitable for longer time scales. We conclude Section 4.5 with the integration of the annual $FCO_2$ cycle, also expressed in mol m$^{-2}$ yr$^{-1}$. Considering the surface area covered by the ES-SOOP-CanOA line (14,000 km$^2$ in the Alboran Sea and 40,000 km$^2$ along the eastern Iberian coast), we then convert the fluxes into Tg $CO_2$ yr$^{-1}$ (a common conversion in the literature). This final estimate provides a high-value result that allows us to determine that this specific ocean region has acted as a net annual atmospheric $CO_2$ sink, and to quantify the magnitude of this uptake.

Can't understand which finding contributed to the assessment made in 2005 by Borges and 2013 by Chen? If these values are from the 2005 and 2013 studies are they still valid ?

The inclusion of those references in the final line of Section 4.5 was an oversight that has now been corrected: the references have been removed. What we intended to convey is that the results of this study contribute to improving our understanding of air-sea $CO_2$ exchange and the role of coastal regions in this process.

**Conclusion**

The section is relatively long and will be beneficial if it's shortened. Especially since parts of this section fit better in the discussion.

We have revised the conclusion by shortening it, removing parts that were repetitive with the discussion, and improving the linkage between the main findings of the study. The revised conclusion is presented below:

**5. Conclusion**

[revised manuscript text omitted]

The part starting in line 896 repeats what is mentioned in line 872.

We modified the *Conclusion* as shown in the response above.

Line 934: The sentence starting "This research…" doesn't make sense.

We modified the *Conclusion* as shown in the response above.

---

## Author Response (AR2)

**Point-by-point response**

Reply to response on multicollinearity in Equation 6: The author's response is understandable, but even if nTA is used, it is still completely dependent on SSS, since TA calculated with SSS is only corrected with SSS. Note that Takahashi et al., 2014 calculated TA + nitrate with SSS, so it is not fully dependent. Such non-independence among variables may lead to instability in each explanatory terms, even if the objective variable (pCO2) is consistent with measured values. At least the Taylor expansion using the three parameters 1: SST, 2: DIC (nDIC) and 3: SSS or TA (nTA) should be estimated to check if the values of the term of SST and DIC are not significantly different from the four-parameter results.

We sincerely appreciate your insightful comment, which has helped us to delve deeper into the calculations and explore new perspectives. We fully understand that the procedure used to reconstruct $A_T$ at the time of each observation may introduce inconsistencies in the Taylor expansion, as it relies on these estimated $A_T$ values. As you point out, the issue stems from the expected multicollinearity between SSS and $A_T$. Given their strong correlation, the Taylor expansion may attribute seasonal changes in $f\text{CO}_{2,sw}$ redundantly or ambiguously, making it difficult to distinguish which part of the signal is strictly due to changes in salinity and which to $A_T$. Furthermore, the individual contributions of $A_T$ or SSS could be biased, potentially leading to over- or underestimations.

While the Taylor expansion assumes statistical independence among the terms (SST, SSS, $A_T$, and $C_T$), using $A_T$ (or $NA_T$) estimated from SSS inherently removes the statistical independence between $A_T$ and SSS. However, it is important to note that the rest of the terms (SST and $C_T$) remain statistically independent and are not affected by this issue of bias or multicollinearity. Considering that the combined contribution of SST and $C_T$ to the seasonal variability of $f\text{CO}_{2,sw}$ exceeds 80% (and even reaches up to 90% in some locations along our transect) the instability caused by the AT–SSS correlation would affect less than 20% of the reconstruction of the seasonal variability of $f\text{CO}_{2,sw}$

Your suggestion has been extremely valuable in helping us improve the robustness of our analysis and increase the certainty in our assessment of the mechanisms driving the seasonal variability of $f\text{CO}_{2,sw}$. In the revised version of the manuscript, we have refined the method used to estimate $A_T$ from SSS. Specifically, we included a non-conservative term ($\varepsilon$) in the linear $A_T$-SSS relationship, which represents the residual (the difference between measured $A_T$ values and those predicted from SSS) and captures the variability not explained by salinity. This $\varepsilon$ term, along with the linear model parameters (slope and intercept) and their associated uncertainties, was propagated through our dataset using Monte Carlo simulations. As a result, the reconstructed $A_T$ values are largely driven by salinity-related processes but also incorporate variability from non-conservative processes that are independent of SSS.

This methodological improvement is now described in detail in the new subsection 2.4.1.2, entitled *"AT determination and reconstruction"* (line 284). These updated $A_T$ values are normalized to reference salinity following the approach of Friis et al. (2003), as we now explain in lines 397-399. The new $A_T$ values were subsequently used to compute $C_T$ with CO2SYS, and we note that the resulting $C_T$ values show only minimal differences (<0.5 µmol kg$^{-1}$) compared to those in the previous version of the manuscript.

We have recalculated all terms involved in the Taylor expansion and included an updated Figure 5 in the revised manuscript. Additionally, in this point-by-point response document, we provide a supplementary figure ("Figure 5 – aux"), which mirrors the structure of Figure 5 and shows a direct comparison between the previously reported and updated values of the $df\text{CO}_2{}^X$.

We confirmed that there is no difference in $\mathrm{d}f\mathrm{CO_2}^{\mathrm{SST}}$ between the previous and revised calculations. The differences in $\mathrm{d}f\mathrm{CO_2}^{\mathrm{CT}}$ and $\mathrm{d}f\mathrm{CO_2}^{\mathrm{SSS}}$ are negligible. These small changes resulted from updates in the $A_T$ input values used in $CO_{2SYS}$, which in turn resulted in slight modifications to the outputted $C_T$ values. For $\mathrm{d}f\mathrm{CO_2}^{\mathrm{AT}}$, we observed differences of less than 2.6 µatm in the S section and less than 7.7 µatm in the E section. These changes correspond to a variation in the relative contribution of $A_T$ to $\mathrm{d}f\mathrm{CO_2}$ of approximately 1-6% compared to the values reported in the previous version.

[Figure]

Figure 5 aux. Temporal evolution of the seasonal rates of $f\mathrm{CO}_{2,sw}$ explained by each of its drivers within the five years of observation. The differences between monthly average data for February and September (where minimum and maximum SST and $f\mathrm{CO}_{2,sw}$ were encountered) was considered to compute the seasonal trends. The standard deviation of the monthly average data was considered in the calculation of the seasonal changes and infers errors in the computation of $f\mathrm{CO}_{2,sw}$, which are summarized in Table Sup3. The dashed purple, blue and green lines represent the $\mathrm{d}f\mathrm{CO_2}^{\mathrm{AT}}$, $\mathrm{d}f\mathrm{CO_2}^{\mathrm{CT}}$ and $\mathrm{d}f\mathrm{CO_2}^{\mathrm{SSS}}$, respectively, of the previos version of the manuscript.

Line 27
Isn't 0.97 -0.97?

It was a typo and was corrected in the revised version.

Line 305
Why do you conclude that nutrient-rich seawater influx reduced the influence of biological activity on TA?

We included that statement more as a hypothesis than as a conclusion, based on the existing literature regarding the inflow of Atlantic water into the Mediterranean. However, we acknowledge that it may be somewhat controversial. We have therefore removed the statement and revised the corresponding paragraphs in the methodology section related to the explanation of the AT reconstruction (see new subsection 2.4.1.2 starting on line 284).

Line 390
Bulk equation for air-sea flux was devised much earlier than 1983 (e.g., Whitmann, 1923; Liss and Slater, 1974). The bulk equation itself is a well-known theory, so there is no problem in omitting its citation.

Citation was removed